

# Frictional properties and microstructural evolution of dry and wet calcite-dolomite gouges

Matteo Demurtas[1], Steven A.F. Smith[2], Elena Spagnuolo[3], and Giulio Di Toro[3,4]

[1]Physics of Geological Processes, The Njord Centre, Department of Geosciences, University of Oslo, Oslo, Norway
[2]Department of Geology, University of Otago, Dunedin 9054, New Zealand
[3]Istituto Nazionale di Geofisica e Vulcanologia (INGV), Rome 00143, Italy
[4]Dipartimento di Geoscienze, Università degli Studi di Padova, 35131 Padova, Italy

**Correspondence:** Matteo Demurtas (matteodemu@gmail.com)

**Abstract.** Calcite and dolomite are the two most common minerals in carbonate-bearing faults and shear zones. Motivated by field examples from exhumed seismogenic faults in the Italian Central Apennines, we investigated the frictional and microstructural evolution of gouge mixtures consisting of 50 wt.% calcite and 50 wt.% dolomite using a rotary-shear apparatus. The gouges were sheared at a range of slip rates (30 $\mu$ms$^{-1}$ – 1 ms$^{-1}$), displacements (0.05–0.4 m), and normal loads (17.5–26

MPa), under both room humidity and water-dampened conditions. The frictional behaviour and microstructural evolution of the gouges were strongly influenced by the presence of water. At room humidity, slip strengthening behaviour was observed up to slip rates of 0.01 ms$^{-1}$, which was associated with gouge dilation and the development of a 500-900 $\mu$m wide slip zone cut by Y-, R-, and R$_1$-shear bands. Above a slip rate of 0.1 ms$^{-1}$, dynamic weakening accompanied the development of a localised <100 $\mu$m thick principal slip zone preserving microstructural evidence for calcite recrystallization and dolomite de-

carbonation, while the bulk gouges developed a well-defined foliation consisting of organized domains of heavily fractured calcite and dolomite. In water-dampened conditions, evidence of gouge fluidization within a fine-grained principal slip zone was observed at a wide range of slip-rates from 30 $\mu$ms$^{-1}$ to 0.1 ms$^{-1}$, suggesting that caution is needed when relating fluidization textures to seismic slip in natural fault zones. Dynamic weakening in water-dampened conditions was observed at 1 ms$^{-1}$, where the principal slip zone was characterised by patches of recrystallized calcite. However, local fragmentation and

reworking of recrystallized calcite suggests a cyclic process involving formation and destruction of a heterogeneous slip zone. Our microstructural data show that development of a well-defined gouge foliation at these experimental conditions is limited to high-velocity (>0.1 ms$^{-1}$) and room humidity, supporting the notion that some foliated gouges and cataclasites may form during seismic slip in natural carbonate-bearing faults.

## 1 Introduction

Calcite and dolomite are the most common minerals in carbonate-bearing faults and shear zones (e.g. Busch and van der Pluijm, 1995; Snoke et al., 1998; Bestmann et al., 2000; De Paola et al., 2006; Molli et al., 2010; Tesei et al., 2014; Fondriest et al., 2015, 2020; Delle Piane et al., 2017). In many cases, the distribution and timing of dolomitization plays an important role in controlling where strain localization occurs. For example, ductile deformation along the Naukluft Nappe Complex in





central Namibia was distributed within a sequence of calcite mylonites, but the main Naukluft Fault formed within heavily
dolomitized layers (Viola et al., 2006; Miller et al., 2008).

Although similar in composition, the rheology, deformation mechanisms, and frictional behaviour of calcite and dolomite
show important differences. Calcite has been widely investigated using microstructural analysis (Kennedy and Logan, 1997;
Kennedy and White, 2001; Liu et al., 2002; Bestmann et al., 2006; Molli et al., 2011) and laboratory experiments over a wide
range of deformation conditions, which includes experiments performed at relatively low strain rates, high temperatures, and
high pressures (e.g Rutter, 1972; Schmid et al., 1980, 1987; De Bresser and Spiers, 1990; Rutter, 1995; Paterson and Olgaard,
2000), and experiments performed at relatively high strain rates, low temperatures, and low pressures (Smith et al., 2013,
2015; Verberne et al., 2014; De Paola et al., 2015; Rempe et al., 2017; Tesei et al., 2017). Comparatively, the rheology and
frictional behaviour of dolomite are relatively poorly understood (e.g. Barber et al., 1981; Weeks and Tullis, 1985). Recently,
the importance of dolomite as a fault and shear zone material in sedimentary and metamorphic settings has been emphasized
in a number of experimental studies (Austin and Kennedy, 2005; Delle Piane et al., 2007, 2008; Davis et al., 2008; De Paola
et al., 2011a, b; Fondriest et al., 2013; Holyoke et al., 2014). At low strain rates, dolomite is brittle up to c. 700 °C (Kushnir
et al., 2015), while calcite can deform plastically at temperatures as low as 150–200 °C (Kennedy and White, 2001). This
pronounced difference in deformation style under similar ambient conditions may significantly influence the rheology of faults
and shear zones in which the two phases co-exist. For example, the occurrence of patches or dispersed grains of dolomite
within calcite-mylonites can result in more efficient strain localization (Oesterling et al., 2007; Kushnir et al., 2015).

Only a few experimental studies have investigated the mechanical behaviour and microstructural evolution of calcite-
dolomite mixtures. Experiments have been performed to study the rheological behaviour of mixtures at relatively high pressures
and temperatures (e.g. torsion experiments of Delle Piane et al., 2009; Kushnir et al., 2015), as well as the frictional behaviour
of mixtures at room temperature over a wide range of strain rates (e.g. room temperature rotary-shear experiments: Mitchell et
al., 2015; Smith et al., 2017; Demurtas et al., 2019a, b). In torsion experiments (confining pressures up to 300 MPa, tempera-
tures of 700–800 °C, shear strain rate $\dot{\gamma}$ = 1-3 × $10^{-4}$ $s^{-1}$, and finite shear strain $\gamma$ < 11), minor quantities of dolomite (e.g. 25
wt.%) in a sintered calcite-rich sample significantly increased the yield strength with respect to pure calcite samples (Kushnir
et al., 2015). Under such experimental conditions, two main deformation mechanisms were observed: brittle fracturing in the
dolomite grains and ductile flow in calcite, possibly as a result of grain boundary sliding assisted by diffusion creep and dislo-
cation glide (Kushnir et al., 2015). Strain hardening observed in these experiments was interpreted to be due to dolomite grains
interrupting more continuous calcite-rich layers and acting as stress concentrators. Brittle failure of dolomite grains eventually
allowed the calcite-rich layers to become continuous and to continue deforming by superplastic flow (Kushnir et al., 2015).

Mitchell et al. (2015) and Smith et al. (2017) studied the frictional behaviour and microstructural evolution of gouge mix-
tures (50 wt.% dolomite and 50 wt.% calcite) deformed at low normal stresses ($\sigma_n \leq 17.5$ MPa), high slip rates (V $\geq$ 0.01
$ms^{-1}$), and large displacements (d = 0.03-3 m), with the aim of reproducing conditions encountered at the base of fast-moving
landslides and during the seismic cycle in shallow-crustal faults. At a slip velocity of 1 $ms^{-1}$, dynamic weakening was associ-
ated with grain size reduction and decarbonation of dolomite within the experimental principal slip zone and in the nearby bulk
gouge. During the early stages of these high-velocity experiments, and prior to the onset of dynamic weakening, a well-defined





foliation developed within the gouge mixtures due to brittle fracturing of calcite and dolomite accompanied by shearing of the
fractured grains in to compositional bands (Mitchell et al., 2015; Smith et al., 2017). These observations indicate that some
natural examples of foliated gouges and cataclasites could form during coseismic shearing (Smith et al., 2017), challenging the
common interpretation that fault rock foliations result from slow aseismic creep (e.g. Rutter et al., 1986; Chester and Chester,
1998; Lin, 2001; Jefferies et al., 2006). Additionally, Demurtas et al. (2019a) documented the presence of a well-defined crys-
tallographic preferred orientation (CPO) in calcite-dolomite gouges, and interpreted the CPO to result from "brittle" processes
involving grain rotation and preferential fracturing along calcite cleavage planes during granular flow at room temperature.
Instead, in regions of the mixed gouge layers that experienced substantial frictional heating during high-velocity slip (i.e.
within the principal slip zone), Transmission Kikuchi Diffraction analysis suggested that nanogranular aggregates deformed
by a combination of grain size sensitive (grains <800 nm) and grain size insensitive (grains >800 nm) plastic creep (Demurtas
et al., 2019b).

The experimental work summarized above indicates that a diverse range of microstructures can form in calcite-dolomite
gouges as a result of both brittle and plastic processes, and that the prevailing microstructures depend on ambient conditions,
strain history, and proximity to zones of shear localization and heating. Potentially, this range of microstructures could be
recognized in natural gouges and cataclasites, which would provide important insights in to the evolution of slip conditions
during the seismic cycle in carbonate-bearing faults. However, to successfully apply the experimental findings to natural fault
zones, a more complete picture of microstructural diversity and its dependence on deformation conditions is required. In
this context, the aim of this paper is to provide (i) a more comprehensive description of the frictional and microstructural
evolution of mixed calcite-dolomite gouges deformed at sub-seismic to seismic slip rates, and (ii) an updated framework for
the interpretation of microstructures found in natural calcite- and dolomite-bearing faults.

## 2   Methods

### 2.1   Starting materials


Synthetic gouges were prepared by mixing 50 wt.% calcite and 50 wt.% dolomite as previously described in Demurtas et al.
(2019a). The calcite-dolomite ratio in the experimental mixtures is similar to that found in natural fault gouges and catacl-
asites from the Vado di Corno Fault Zone (VCFZ, Italian Central Apennines; Demurtas et al., 2016), and used in previous
experimental studies (Smith et al., 2017). The calcite gouge was derived by crushing Carrara marble with a modal composition
of 98.8 wt.% calcite and <1 wt.% dolomite and muscovite (see Supplementary Material). The dolomite gouge was derived by
crushing dolomitized portions of the Calcare Massiccio Formation from the VCFZ (Demurtas et al., 2016). The crushed gouges
were passed through a 250 $\mu$m sieve and then mixed together by slow tumbling for c. 30 minutes. Two batches of gouge were
prepared (CDM1, calcite = 47.2 wt.% and dolomite = 52.8 wt.%; CDM2, calcite = 42.9 wt.% and dolomite = 57.1 wt.%).



## 2.2 Experimental setup and deformation conditions

Nineteen experiments were performed at slip rates from 30 $\mu$ms$^{-1}$ to 1 ms$^{-1}$ with SHIVA (Slow- to HIgh-Velocity rotary-shear friction Apparatus) at the Istituto Nazionale di Geofisica e Vulcanologia in Rome (Di Toro et al., 2010; Niemeijer et al., 2011) (Table 1). The gouges were deformed inside a metal holder specifically designed for incohesive materials (Fig. 1; Smith et al., 2013, 2015). The thickness of the gouge layers at the start of the experiments was c. 3 mm. Horizontal displacements of the axial column were sampled at 2.5 Hz–25 kHz, and measured using a direct current differential transformer (DCDT, 50 mm range and c. 50 $\mu$m resolution) and a linear variable differential transformer (LVDT, 3 mm range and c. 0.03 $\mu$m resolution).

Further details of the data acquisition system, and location and calibration of the load cells, detectors, and devices, are found in Niemeijer et al. (2011) and Smith et al. (2013).

Measurements of room humidity and room temperature were collected at a distance of <1 cm from the gouge holder before and during the experiments (Fig. 1a). Temperature variations during deformation were measured at an acquisition rate of 2.5

Hz using four K-type thermocouples (Nickel-Alumel) installed on the stationary side of the gouge holder (Fig. 1a-b; Demurtas et al., 2019a). One thermocouple was positioned at c. 200 $\mu$m from the gouge layer (Fig. 1b). The other three thermocouples were located in the sample holder and stationary column to detect temperature variations due to heat conduction through the gouge holder and apparatus (Fig. 1a). $CO_2$ emissions were monitored using an OmniStar™ GSD 301 O mass spectrometer designed for gas analysis at atmospheric pressure.

Experiments were performed at both room humidity and water-dampened conditions with normal stress held constant at 17.5±0.1 MPa, with the exception of experiment *s1324* performed at 26 MPa normal stress (Table 1). Room humidity varied between 41% and 62%, and the room temperature between 19 °C and 22 °C. In water-dampened conditions, c. 2 ml of deionized water was added to the top of the gouge layer using a pipette before the gouge holder was positioned in the apparatus. Experiment *s1327* was performed using a specially designed water bath that ensured saturation within the gouge layer dur-

ing this long-duration experiment (Supplementary Material). Experiments were performed at target slip rates ranging from 30 $\mu$ms$^{-1}$ to 1 ms$^{-1}$, with acceleration and deceleration of 6 ms$^{-2}$. Total displacements ranged from 0.05 m to 0.4 m. Two compaction experiments were performed by applying a normal stress of 17.5 MPa for 300 s (i.e. static load experiments in Table 1) and used as references for the microstructure of the starting materials.

## 2.3 Analytical techniques

After each experiment, the entire gouge layer was recovered and impregnated in low-viscosity epoxy (Araldite 2020) for microstructural analysis. Polished thin sections were cut perpendicular to the gouge layer and subparallel (i.e. tangential cut) to the slip direction (Fig. 1d). Microstructural analysis was performed with a Zeiss Sigma VP Field-Emission Scanning Electron Microscope (SEM) at the Otago Micro and Nanoscale Imaging facility (OMNI; University of Otago. Acquisition conditions for backscattered electron images: accelerating voltage 15 kV, working distance 6-7 mm). Energy-dispersive X-ray spectroscopy

(EDS) in the SEM was used to produce element maps showing the distribution of calcium and magnesium. Crystallographic orientation data from calcite were acquired by electron backscatter diffraction (EBSD) on SYTON-polished thin sections.





Data were collected with a NordlysF EBSD camera from Oxford Instruments and processed using AZtec software (Oxford Instruments). Mineralogical changes that occurred during the experiments were determined by semi-quantitative X-ray powder diffraction (XRPD) conducted in the Department of Geoscience, University of Padova. The XRPD analyses were performed
on both the bulk gouges and on small intact chips of the localized slip surfaces that formed in the experiments.

## 3 Results

### 3.1 Friction evolution with slip and slip rate

The evolution of the effective friction coefficient ($\mu$) with slip and slip rate was influenced by the availability of water during deformation (Figs. 2, 3). In room humidity conditions and slip rates $\leq 0.01$ ms$^{-1}$, the calcite-dolomite mixtures showed a pro-
gressive increase of $\mu$ (slip strengthening behaviour) up to 0.75-0.80 (measured between 0.15 m and 0.35 m of slip) following an initial peak fiction ($\mu_{peak}$) of 0.64-0.71 (Figs. 2, 3). At a slip rate of 0.1 ms$^{-1}$, a substantial decrease of $\mu$ was observed (slip weakening behaviour to steady state $\mu_{ss}$ of 0.55±0.01) following a prolonged initial strengthening phase (c. 0.062 m) that reached $\mu_{peak}$ of 0.68 (Fig. 2b). Significant dynamic weakening was observed at a slip rate of 1 ms$^{-1}$ at 17.5 MPa and 26 MPa normal stress (experiments *s1221* and *s1324*, respectively), following a short initial strengthening phase (lasting c. 0.005-0.008
m) that was followed by a steady state of $\mu_{ss}$ of 0.25-0.28 (Figs. 2a, 3b). In these two experiments, a re-strengthening phase (final $\mu$ up to c. 0.56-0.59) was observed during deceleration of the rotary column.

In water-dampened conditions, the gouge mixtures showed a similar evolution of friction at slip rates $\leq 0.1$ ms$^{-1}$, characterized by slight slip strengthening to slip neutral behaviour (Fig. 2c). Notably, $\mu_{peak}$ and $\mu_{ss}$ were lower than in room humidity experiments, with $\mu_{peak}$ = 0.61-0.64 and $\mu_{ss}$ = 0.62-0.70 (Fig. 3). At a slip rate of 1 ms$^{-1}$, the initial strengthening
phase was much shorter than in room-humidity conditions (c. 0.003 m), and dynamic weakening resulted in $\mu_{ss}$ of 0.31±0.02. Re-strengthening was also observed during deceleration, with an increase in $\mu$ up to 0.57.

### 3.2 Gouge thickness evolution with slip rate

No significant gouge loss was observed during the experiments, with the exception of those performed at V = 0.1 ms$^{-1}$ discussed below. Therefore, the evolution of axial displacement is interpreted to result from changes in gouge layer thickness due to
dilation and compaction. In room humidity conditions, the evolution of gouge layer thickness depends on slip rate (Fig. 4a). At V $\leq$ 0.001 ms$^{-1}$, the gouge layers show a three-stage evolution: (i) initial compaction of c. 90-120 $\mu$m at the onset of sliding, (ii) dilation of c. 50-70 $\mu$m during the slip strengthening phase, and (iii) approximately constant thickness once the steady state friction coefficient is reached. Overall compaction of c. 30-60 $\mu$m is recorded. At V = 0.01 ms$^{-1}$, initial compaction of 100 $\mu$m is followed by approximately constant thickness (Fig. 4a). At higher slip rates (V $\geq$ 0.1 ms$^{-1}$), continuous compaction was
observed throughout the experiments (up to c. 300 $\mu$m of axial shortening at V = 1 ms$^{-1}$), and compaction rate increased with slip rate (Fig. 4a).



Under water-dampened conditions, the gouge mixtures exhibit a similar evolution of thickness irrespective of slip rate (Fig. 4b). Compaction was initially rapid in the first few cm of sliding, and then reached an approximately constant compaction rate that was similar in all experiments. Total compaction of c. 200-250 $\mu$m was recorded (Fig. 4b).

## 3.3 Temperature evolution and $CO_2$ emissions

Figure 5a shows maximum temperatures measured by the thermocouple located closest to the gouge layers (Fig. 1b; Demurtas et al., 2019a, described temperature evolution with slip). The maximum temperature (621 °C) was achieved in experiment *s1221* performed under room humidity conditions at V = 1 ms$^{-1}$ (Fig. 5a). For the same slip rate and normal stress, but in water-dampened conditions, the maximum temperature was 210 °C (Fig. 5a). Temperature increases were detected in all
experiments at slip rates $\geq$ 0.01 ms$^{-1}$, and the maximum temperature increased with increasing slip rate (Fig. 5a).

$CO_2$ emissions above ambient levels were only detected in experiments at slip rates $\geq$ 0.1 ms$^{-1}$ (Fig. 5b). Because the mass spectrometer was not calibrated and the sample holder was open to the laboratory, the data can only be used in a qualitative way. In room humidity conditions, the intensity of the $CO_2$ peak was significantly higher at 1 ms$^{-1}$ than at 0.1 ms$^{-1}$. In water-dampened conditions, the $CO_2$ peaks were substantially smaller than at equivalent room humidity conditions.

## 3.4 Mineralogy of deformed gouges

Compared to the starting materials, no mineralogical changes were detected in any of the deformed bulk gouges (see Supplementary Material). In room humidity experiment *s1210* (30 $\mu$ms$^{-1}$), a slight broadening of the main peak for calcite was observed (Fig. 6a), and to a lesser degree also for dolomite. XRPD analysis of cohesive chips recovered from the slip surface of water-dampened experiment *s1214* (V = 30 $\mu$ms$^{-1}$) indicates the presence of aragonite (Fig. 6b). At V = 1 ms$^{-1}$ and room humid-
ity conditions, the recovered slip surface was composed of dolomite, Mg-calcite, and periclase (MgO) (Fig. 6c). Mg-calcite and periclase are two of the main products of dolomite decarbonation that starts at c. 550 °C ($MgCa(CO_3)_2 \rightarrow MgO + (Ca, Mg)CO_3 + CO_2$, Samtani et al., 2002; De Paola et al., 2011a, b).

## 3.5 Microstructures of deformed gouge layers

Figure 7 summarizes the range of microstructures that developed in room humidity and water-saturated conditions at different
slip rates. As noted in previous experiments (e.g. Kitajima et al., 2010; Smith et al., 2017), several distinct microstructures and microstructural domains were recognized, defined by variations in grain size, fabric, and the presence of localized slip surfaces. In the present set of experiments, the microstructural domains and the principal slip zone varied in thickness at different slip rates (Fig. 7). In Figure 8, the thickness of the principal slip zone has been tracked at different deformation conditions (i.e. slip rate and presence of water).





### 3.5.1 Microstructures of room humidity experiments

At slip rates $\leq 0.01$ ms$^{-1}$, gouges were characterized by the development of a 500-900 $\mu$m thick slip zone (Fig. 9a), consisting of a fine-grained matrix (grain size c. 1 $\mu$m) containing subrounded grains of dolomite c. 5-10 $\mu$m in size (Fig. 9b). The slip zone contains sub-parallel, 10-30 $\mu$m thick Y, R, and R$_1$ type shear bands (using the terminology of Logan et al., 1979, Fig. 9a,c). Each individual shear band is associated with a very fine-grained matrix (grain size <1 $\mu$m) composed of calcite and dolomite. The presence of multiple interlinked shear bands contributes to a weak foliation within the slip zone that lies sub-parallel to gouge layer boundaries (Fig. 9a). Y-, R-, and most notably R$_1$-shears, gradually decrease in abundance with increasing slip rate. The transition from fine-grained slip zone to highly fractured bulk gouge is typically well-defined (see upper part in Fig. 9d). The bulk gouge shows widespread cataclasis and intragranular fracturing, which is focussed preferentially into calcite grains (Smith et al., 2017; Demurtas et al., 2019a). Fractures that cut relatively large grains of calcite in the bulk gouge often exploit cleavage planes (e.g. Fig. 9d; Smith et al., 2017; Demurtas et al., 2019a).

At a slip rate of 0.1 ms$^{-1}$, the bulk gouge develops a weak foliation defined by compositional banding of heavily fractured calcite- and dolomite-rich domains, which lie adjacent to a localized principal slip zone c. 110 $\mu$m thick (Fig. 9e). The foliation is inclined 25-30° to the principal slip surface and appears to form by disaggregation and shearing of originally intact calcite and dolomite grains (Fig. 9e). Locally, the principal slip surface is associated with discontinuous lens-shaped patches (up to 15-20 $\mu$m thick) of calcite with irregular boundaries and negligible porosity (Fig. 9f).

In experiments conducted at 1 ms$^{-1}$, the bulk gouges developed a well-defined foliation across most of the thickness of the layers (Smith et al., 2017; Demurtas et al., 2019a, b). The foliation is defined by alternating calcite- and dolomite-rich domains inclined at c. 40° to the principal slip surface (Fig. 10a-b), which become progressively rotated as they approach the slip surface (Fig. 10c). Large remnant grains (up to 200 $\mu$m) in the bulk gouge are often rimmed by fractured tails of finer-grained aggregates (grain size <10 $\mu$m), and resemble mantled porphyroclasts in mylonites (e.g. Snoke et al., 1998; Trouw and Passchier, 2009) (arrow in Fig. 10a). At distances of <400 $\mu$m from the principal slip surface, the mean grain size decreases substantially, there are very few large surviving grains (up to c. 100 $\mu$m in size), and there is a greater degree of mixing between calcite and dolomite (see more uniform colouring in the upper part of EDS map in Fig. 10b). The principal slip zone consists of a 15-20 $\mu$m thick, extremely fine-grained layer ($\ll$1 $\mu$m in size) composed of calcite, Mg-calcite, dolomite, and periclase (EDS and XRPD analysis; Figs. 6 and 10c-d). Calcite forms elongate aggregates with negligible porosity that display an aggregate preferred orientation with the long axes sub-parallel to foliation (Fig. 10c-e). Dolomite-rich domains show higher porosity and preserve distinct grain structures (Fig. 10c-d). EBSD analysis of elongate calcite aggregates within the principal slip zone (Fig. 10e) shows a distinct crystallographic preferred orientation with c-axes inclined sub-perpendicular to gouge layer boundaries (Fig. 10f; see also Demurtas et al., 2019b). Adjacent to the principal slip zone, a c. 30-40 $\mu$m thick layer includes dolomite grains with diffuse internal cracking, clusters of small holes, and vesicular rims previously interpreted as resulting from degassing during decarbonation of comminuted dolomite grains (Fig. 10c; Mitchell et al., 2015; Demurtas et al., 2019b). At 1 ms$^{-1}$ and 26 MPa, the foliation was found only within 400 $\mu$m of the principal slip surface (Supplementary





Material). The principal slip zone was composed of a calcite-rich recrystallized layer, with substantially reduced porosity and well-rounded dolomite clasts a few micrometres in size (Supplementary Material).

### 3.5.2 Microstructures of water-dampened experiments

In the bulk gouges, the region furthest from the slip zone is composed of grains that show very limited fracturing and resemble the starting materials (Fig. 11a,d; compare with Fig. 1e). Towards the slip zone, grains are increasingly fractured and become rounder. As in the room humidity experiments, most of the larger "surviving" grains are composed of dolomite (Fig. 11e), consistent with data showing that calcite undergoes more efficient grain size reduction compared to dolomite (Smith et al., 2017; Demurtas et al., 2019a). Domain boundaries (e.g. between intact bulk gouge and comminuted gouge) are often gradational (Fig. 11d), and the total thickness of the comminuted zone is observed to decrease at higher slip rates (from c. 1500 $\mu$m thick at 30 $\mu$ms$^{-1}$ to c. 150 $\mu$m thick at 1 ms$^{-1}$). The principal slip zone consists of an ultrafine-grained matrix (grain size <1 $\mu$m) composed of a mixture of calcite and dolomite, with a few well-rounded surviving dolomite grains up to 20-30 $\mu$m in size (Fig. 11b-c). At the lowest slip rate (i.e. 30 $\mu$ms$^{-1}$), the principal slip zone has a sharp boundary with a characteristic wavelength with the underlying gouge (Fig. 11d), and contains irregular flame-like structures defined by subtle variations in the content of calcite and dolomite (Fig. 11b). The principal slip zone is cut by discrete slip surfaces oriented subparallel to the boundaries of the gouge (Y-shear, Fig. 11c). Experiments performed at 30 $\mu$ms$^{-1}$ with increasing displacement (*s1327*, *s1329*, *s1328*, *s1330-s1214* with displacement of 0.05-0.1-0.2-0.4 m, respectively; Table 1) show that the three distinct microstructural domains are already recognizable after <0.05 m of slip (Fig. 11d), and that the final thickness of each microstructural domain is a function of total slip and slip rate. At a slip rate of 0.1 ms$^{-1}$, the comminuted principal slip zone shows distinct grain-size grading, characterized by an abundance of relatively large and angular dolomite particles towards the stationary side of the slip zone, and an absence of such particles towards the rotary side (Fig. 11e). Measurements of the thickness of the principal slip zone at different slip rates (Fig. 8) show a log-linear decrease in thickness from c. 400 $\mu$m at 30 $\mu$ms$^{-1}$ to c. 30 $\mu$m at 1 ms$^{-1}$.

At a slip rate of 1 ms$^{-1}$, the gouge contains an intensely comminuted c. 300-400 $\mu$m thick layer bordering the principal slip zone (Fig. 11f). The transition between the two domains is sharp and characterized in places by the occurrence of discrete Y-shears. The principal slip zone consists of lens-shaped patches of a calcite-rich and fine-grained (grain size <1 $\mu$m) layer c. 30 $\mu$m thick with negligible porosity, which is embedded in a highly comminuted and fine-grained matrix containing a few larger dolomite grains. The principal slip surface cuts sharply through this layer and truncates larger clasts (Fig. 11f-g). Locally, reworked angular fragments of the principal slip zone are found (Fig. 11h).

## 4 Discussion

### 4.1 Microstructural evolution and weakening mechanisms in calcite-dolomite mixtures

The mechanical behaviour and microstructural evolution of calcite-dolomite gouges show substantial differences based on the availability of water during deformation. In room humidity conditions, slip strengthening at slip rates ≤ 0.01 ms$^{-1}$ is associated





with (i) initial compaction followed by dilation (Fig. 4a) and (ii) the development of a >500 $\mu$m thick slip zone composed of a
245    fine-grained (c. 1 $\mu$m) calcite-dolomite mixture cut by Y-, R-, and $R_1$-shear bands (Fig. 9). These observations have previously
been interpreted to relate to the development and broadening of a distributed zone of deformation during strain hardening (e.g.
Marone et al., 1990; Beeler et al., 1996; Rathbun and Marone, 2010). This is also supported by widening of the main peaks
for calcite and dolomite in XRPD analysis (Fig. 6a), which is interpreted to result from a decrease in the mean crystallite size
or deformation-induced microstrain within the crystallites (e.g. Ungár, 2004). A much shorter initial period of dilatancy is
250    observed in experiments performed at V $\geq$ 0.1 ms$^{-1}$ (Fig. 4a), and this correlates with (i) a transition from slip hardening to
slip weakening (Figs. 2-3) and (ii) the development of a well-defined, localized principal slip zone that accommodates most of
the strain after it forms (Figs. 9e-f and 10) (see also Han et al., 2007; Fondriest et al., 2013; Smith et al., 2013, 2015; Green
et al., 2015; De Paola et al., 2015; Mitchell et al., 2015; Rempe et al., 2017; Pozzi et al., 2019; Demurtas et al., 2019b). The
switch to slip weakening friction and a higher degree of strain localization is also accompanied by a significant temperature
255    rise generated within the principal slip zone (Fig. 5a), $CO_2$ emissions (Fig. 5b), and the formation of Mg-calcite and periclase
in samples collected from the principal slip zone (Fig. 6c). Collectively, these observations suggest that the temperature rise at
relatively high slip velocities caused dynamic weakening and decarbonation of dolomite (and possibly calcite).

At high slip rates (V $\geq$ 0.1 ms$^{-1}$), the onset of dynamic weakening in carbonate gouges deformed at room humidity has
been interpreted as a consequence of local heating along incipient slip surfaces, which eventually coalesce into a localized
260    and through-going shear band (De Paola et al., 2015; Smith et al., 2015; Rempe et al., 2017). Further slip then increases
the bulk temperature due to continued frictional heating in the principal slip zone and dissipation of heat in to the bulk gouge,
resulting in local gouge recrystallization (Smith et al., 2015). Under these conditions, high strain rates can be accommodated by
temperature- and grain size-dependent deformation mechanisms leading to "viscous" flow (Green et al., 2015; De Paola et al.,
2015; Pozzi et al., 2018, 2019; Demurtas et al., 2019b). Demurtas et al. (2019b) performed Transmission Kikuchi Diffraction
265    (TKD) analysis on electron-transparent samples of the low porosity, fine-grained principal slip zone of experiment *s1221* (V =
1 ms$^{-1}$ under room humidity conditions; Fig. 10c) to investigate the deformation mechanisms active during coseismic sliding in
calcite-dolomite mixtures. Their results show that the principal slip zone is composed of a nanogranular aggregate made of two
grain populations: (i) nanograins 100-300 nm in size exhibiting low internal lattice distortion, compatible with deformation
by grain size sensitive creep, and (ii) nanograins >800 nm in size showing development of subgrains, suggesting deformation
by grain size insensitive creep (Demurtas et al., 2019b). Although the maximum temperature measured during deformation
in the present experiments was 621 °C (Fig. 5a), accommodation of the calculated strain rates ($\dot{\gamma} = 6 \times 10^3$ s$^{-1}$) could be
explained by the significant decrease of the activation energy for creep mechanisms (but also decarbonation reactions) due to
the nanogranular nature of the particles (Demurtas et al., 2019b). Similar observations have been documented by Pozzi et al.
(2019) in experimental nanogranular principal slip zones in pure calcite gouges deformed at coseismic slip rates. Alternatively,
the temperatures achieved in the slip zones of high velocity (V = 1 ms$^{-1}$) but short duration experiments (<0.5 s) could be higher
than those measured with thermocouples and estimated using numerical models. The thermocouples used here were located a
few mm from the edge of the slipping zone (Fig. 1b). Additionally, they have large thermal inertia and low real acquisition rates
because the electric potential developing in response to temperature changes is very slow (0.1-0.5 s) compared to the duration





of the experiments (c. 0.5 s) (Sarnes and Schrüfer, 2007). Recent studies in which the temperature during experimental seismic

slip was measured with optical fibres located inside the slip zone (in-situ measurements at acquisition rates of 1 kHz) detected temperatures 300-400 °C higher than those measured with thermocouples (Aretusini et al., 2019). Temperatures in the slipping zone substantially higher than 621 °C would make grain size- and temperature-dependent deformation mechanisms more efficient. Instead, in the case of experiment *s1218* performed at V = 0.1 ms$^{-1}$, the moderate dynamic weakening ($\mu_{ss}$ = 0.55) can be related to more limited frictional heating within the principal slip zone both in time (max temperature measured of

190 °C, Fig. 5a) and space (patchy recrystallized areas in Fig. 9e-f). However, at least locally, the temperature increase was sufficiently large to decompose dolomite, as testified by the clear $CO_2$ peak measured during shearing at this velocity (Fig. 5b).

In water-dampened conditions, the mechanical behaviour of the calcite-dolomite mixtures is similar (slight slip strengthening to slip neutral) at all slip rates up to 0.1 ms$^{-1}$ (Fig. 2c). The thickness of the principal slip zone decreases log-linearly with increasing slip rate, indicating a progressively higher degree of localization (Fig. 8). However, this has no obvious effect on

the steady state friction coefficient (Fig. 3b), possibly suggesting that the steady state is controlled by strain and that strain is kept constant by microstructural reorganization distributed within the slip zone. The principal slip zone is composed of a very fine-grained ($\ll 1~\mu$m) matrix of calcite and dolomite that includes a few well-rounded dolomite clasts up to 20-30 $\mu$m in size (Fig. 11). The similarity in microstructure at all investigated slip rates suggests that water has a major role in promoting faster grain size reduction at the onset of slip, possibly by decreasing the surface energy and yield stress of calcite and dolomite

(Risnes et al., 2005; Røyne et al., 2011). XRPD analysis of the slip surface of experiment *s1214* (30 $\mu$ms$^{-1}$) showed the formation of aragonite (Fig. 6b). Given that the starting materials were composed of calcite and dolomite only, the aragonite must have formed during deformation. Li et al. (2014), documented polymorphic transformation of calcite into aragonite due to mechanical grinding in a dry (i.e. room humidity) environment. Our observations therefore suggest that relatively dry patches could develop in the gouge layer during slip (or were present at the onset of slip), or that such transformation is also possible

under water saturated conditions.

The slip zone of the water-dampened experiment performed at 30 $\mu$ms$^{-1}$ is characterized by flame-like structures, and domain boundaries that display a characteristic wavelength (Fig. 11b,d). Similar structures are typical of soft sediment deformation (Allen, 1985) and have also been described within fault cores (Brodsky et al., 2009), where they are interpreted to result from fluid mobilization and a difference in viscosity between two adjacent layers during deformation. Additionally, the occurrence

of grain size grading within the water-dampened principal slip zone formed at a slip rate of 0.1 ms$^{-1}$ is indicative of grain rearrangement due to frictional sliding (see Masoch et al., 2019, and reference therein), referred to as the "Brazil nut" effect, a phenomenon observed when large grains move to the top of a fluidized layer due to differences in dispersal pressure between large and small particles (Williams, 1976). Grain size grading was reported by Boullier et al. (2009) from the principal slip zone of the 1999 $M_W$ 7.6 Chi-Chi earthquake in Taiwan, and from exhumed normal faults in Alpine Corsica (Masoch et al.,

2019). Additionally, similar microstructures were produced by Boulton et al. (2017) in high velocity rotary-shear experiments (V = 1 ms$^{-1}$) performed on clay-rich drill chips retrieved from the Alpine Fault (New Zealand). As in the experiments by Boulton et al. (2017), grain size grading in our experiments was observed only in water-dampened conditions. Collectively, the presence of flame-like structures, undulating domain boundaries with a characteristic wavelength, and grain-size grading,



suggests that water-dampened gouges experienced fluidization at slip rates between 30 $\mu$ms$^{-1}$ and 0.1 ms$^{-1}$. This is significant
because textures related to fluidization in natural gouges and cataclasites are often interpreted to form during coseismic slip at
high velocities (e.g. Monzawa and Otsuki, 2003; Rowe et al., 2005; Boullier et al., 2009; Brodsky et al., 2009; Demurtas et
al., 2016; Boulton et al., 2017; Smeraglia et al., 2017). Various mechanisms have been proposed to account for fluidization of
granular materials in fault zones, including (i) frictional heating and thermal pressurization (Boullier et al., 2009), (ii) dilation
that limits grain-grain contacts (Borradaile, 1981; Monzawa and Otsuki, 2003), and (iii) focussed fluid flow along slip zones
during and after coseismic sliding (e.g. fault-valve mechanism of Sibson, 1990). In our experiments, temperature measurements
made during slip at 30 $\mu$ms$^{-1}$ suggest that significant frictional heating is unlikely, and therefore thermal pressurization is an
unlikely mechanism within the slip zone. Water-dampened experiments are characterized by continuous compaction, which
also excludes the dilation-related hypothesis of Borradaile (1981). We propose that fluidization in the principal slip zone might
be caused by local fluid pressure increase within water-saturated patches as a result of continuous compaction combined with
minimal fluid loss during deformation. A sudden release of water from the pressurized patch could result in gouge mobilization
and injection of material in to the adjacent slip zone.

In water-dampened experiments at 1 ms$^{-1}$, abrupt dynamic weakening preceded by a very short-lived strengthening phase has
previously been documented in experiments on calcite gouges (Rempe et al., 2017) and calcite marbles (Violay et al., 2014). In
gouges, Rempe et al. (2017) suggested that the rapid onset of dynamic weakening could be related to faster grain size reduction
in the presence of water, leading to an early switch from brittle deformation to grain size sensitive creep in the principal slip
zone, analogous to the process suggested to occur in dry gouges (De Paola et al., 2015; Demurtas et al., 2019b; Pozzi et al.,
2019). However, there is an apparent discrepancy between the relatively low maximum temperature measured close to the
principal slip zone in water-dampened experiments (200 °C at V = 1 ms$^{-1}$; Fig. 5a), and the observed $CO_2$ production (Fig.
5b) combined with microstructural evidence for recrystallization during deformation (Fig. 11f-g). As previously discussed, this
could be due to an underestimate of peak temperature (Aretusini et al., 2019). Alternatively, Ohl et al. (2020) proposed that
mechanical liming (see Martinelli and Plescia, 2004) along natural faults could be a possible slip weakening mechanism that
does not necessarily involve a macroscopic temperature increase of >500-600 °C.

## 4.2   Foliation development in calcite-dolomite gouges at coseismic slip rates

Foliated gouges and cataclasites are common fault rocks in the brittle upper crust (Snoke et al., 1998). Typically, they are
interpreted to form due to a combination of cataclasis and dissolution-precipitation reactions during aseismic fault creep (e.g.
Rutter et al., 1986; Chester and Chester, 1998; Lin, 2001; Collettini and Holdsworth, 2004; Jefferies et al., 2006; De Paola et
al., 2008; Wallis et al., 2013). Experimental observations support this idea, and show that well-defined foliations can form as a
result of dissolution-precipitation reactions accompanied by granular flow and frictional sliding at low slip rates (V < 1 $\mu$ms$^{-1}$;
Bos et al., 2000; Niemeijer and Spiers, 2006).

However, the association of foliated fault rocks with possible microstructural indicators of seismic slip in natural fault
rocks (e.g. mirror-like slip surfaces with truncated clasts, see Demurtas et al., 2016) led Smith et al. (2017) to investigate the
possibility that some foliated gouges and cataclasites might have a coseismic origin. Rotary-shear experiments performed at





a slip rate of 1.13 ms$^{-1}$ on gouges composed of 50 wt.% calcite and 50 wt.% dolomite showed the development of a foliation defined by an organized banding of heavily fractured calcite and dolomite clasts (Smith et al., 2017). Experiments performed at increasing displacements revealed that the foliations are established during the initial strengthening phase, when distributed strain throughout the bulk gouge causes grain comminution and distributed shearing. Once dynamic weakening occurs, strain progressively localizes into a single continuous principal slip zone, and the foliation in the bulk gouge does not show any further microstructural change. Shear strain analysis of the foliated layers showed that relatively low values of strain ($\gamma < 4$) are needed to develop a foliation. Based on their observations, Smith et al. (2017) suggested that some natural foliated gouges and cataclasites characterized by compositional banding, grain size variations, and preferred particle or fracture alignments, could form by distributed brittle flow as strain localizes during coseismic shearing, especially if such foliations are found in proximity to microstructural indicators of coseismic slip. The experiments presented in this paper allow us to test this hypothesis over a wider range of slip rates (i.e. 30 $\mu$ms$^{-1}$ – 1 ms$^{-1}$) and deformation conditions (i.e. room humidity vs. water dampened). The formation of well-defined foliations throughout the bulk gouge was only observed at high slip rates (V = 1 ms$^{-1}$) and room humidity conditions (Fig. 10), corresponding to the conditions presented in Smith et al. (2017). Local foliation development was also observed in lower velocity experiments at room humidity, although this was restricted to regions <400 $\mu$m from the principal slip surface (Fig. 9e).

Our new experiments support the hypothesis presented in Smith et al. (2017) that well-defined foliations in calcite-dolomite gouges can form during high velocity sliding in carbonate gouges, and could be used in conjunction with other microstructures (e.g. seismic slip indicators) to better understand localization processes in faults. The lowest slip velocity studied here (i.e. 30 $\mu$ms$^{-1}$) is still too high for pressure-solution to be efficient in calcite or dolomite. Lower slip rates might promote the activation of pressure-solution, which could result in the formation of a foliation under certain conditions. Grain elongation and foliation development have previously been reported in experiments performed on calcite-dolomite mixtures by Delle Piane et al. (2009). However, in this case, the deformation conditions (i.e. torsion experiments at temperatures of 700-800 °C, confining pressure of 300 MPa, shear strain rate of $3 \times 10^{-4/-5}$ s$^{-1}$) were representative of mid- to lower crustal depths, rather than the low-pressure low-temperature ambient conditions explored here.

## 5   Conclusions

A series of rotary-shear experiments was performed on gouges composed of 50 wt.% calcite and 50 wt.% dolomite to develop an understanding of microstructural evolution at a range of slip rates (30 $\mu$ms$^{-1}$ – 1 ms$^{-1}$), fluid conditions (room humidity and water dampened), total displacements (0.05–0.4 m), and normal loads (17.5 and 26 MPa).

The evolution of the apparent friction coefficient is strongly influenced by the presence of water: at room humidity, slip strengthening is observed up to slip rates of 0.01 ms$^{-1}$, above which dynamic weakening occurs. In water-dampened conditions, slight slip strengthening to slip neutral friction characterises experiments up to slip velocities of 0.1 ms$^{-1}$, above which dynamic weakening occurs abruptly. The mechanical differences observed under room humidity and water-dampened conditions are also reflected in the microstructures of the deformed gouge layers. At room humidity, slip strengthening is associated with



diffuse deformation and the development of a relatively thick slip zone cut by Y-, R-, and $R_1$-shear bands. The onset of dynamic weakening is concomitant with the development of a localised principal slip zone containing evidence of dolomite decarbonation and calcite recrystallization. In the presence of water, evidence of gouge fluidization within a fine-grained principal slip zone is observed at slip rates from 30 $\mu$ms$^{-1}$ to 0.1 ms$^{-1}$, suggesting that fluidization may not be restricted to

coseismic slip rates. At 1 ms$^{-1}$, the principal slip zone is characterised by patches of recrystallized calcite that are locally broken and reworked.

The development of a well-defined foliation in the bulk gouge layer only occurs in room humidity experiments at a slip rate of 1 ms$^{-1}$, consistent with the work of Smith et al. (2017). This observation supports the notion that some foliated gouges and cataclasites may form during coseismic slip in natural carbonate-bearing faults.

*Data availability.* Mechanical data from the experiments are available upon request from the corresponding author.

*Author contributions.* MD, SAFS, ES and GDT designed the project. MD and ES performed the experiments. MD and SAFS performed the microstructural analysis. MD, SAFS, ES and GDT were part of the discussion and contributed to the writing of the manuscript.

*Competing interests.* The authors declare that they have no conflict of interest.

*Acknowledgements.* MD, ES and GDT were supported by the European Research Council Consolidator grant project 614705 NOFEAR.

SAFS acknowledges the Marsden Fund Council (projects UOO1417 and UOO1829) administered by the Royal Society of New Zealand. Stefano Aretusini and Michele Fondriest are thanked for fruitful discussions. Marianne Negrini provided assistance with the SEM at the Otago Centre for Electron Microscopy, University of Otago. Federico Zorzi is thanked for performing the XRPD analysis and Leonardo Tauro for assistance during thin section preparation.



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



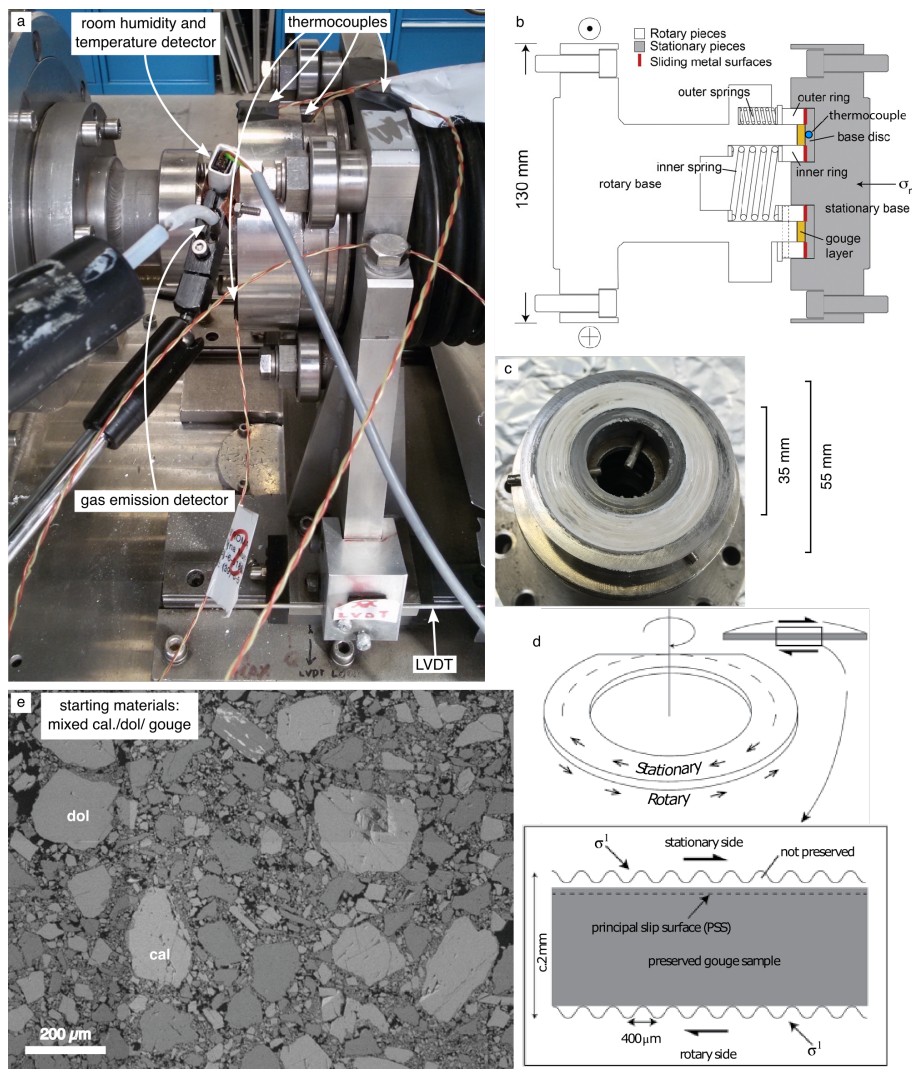

**Figure 1.** Rotary-shear experimental setup. a) Detectors in the sample chamber. Gas emission, humidity, and temperature detectors were placed at <1 cm from the sample holder. Four thermocouples were placed on the stationary side at increasing distances from the gouge layer. Note: the position of the thermocouple nearest to the gouge layer is not visible here and is illustrated in b). b) Diagram of the gouge holder with the location of the thermocouple nearest to the gouge layer (modified after Smith et al., 2015). c) Sample appearance post deformation with mirror-like slip surface formed in an experiment performed at $V = 0.1$ ms$^{-1}$. d) Diagram showing the location of the recovered and analysed gouge layer after the experiment (after Smith et al., 2017). e) SEM backscattered electron (SEM-BSE) image of the starting material after applying 17.5 MPa for 300 s.



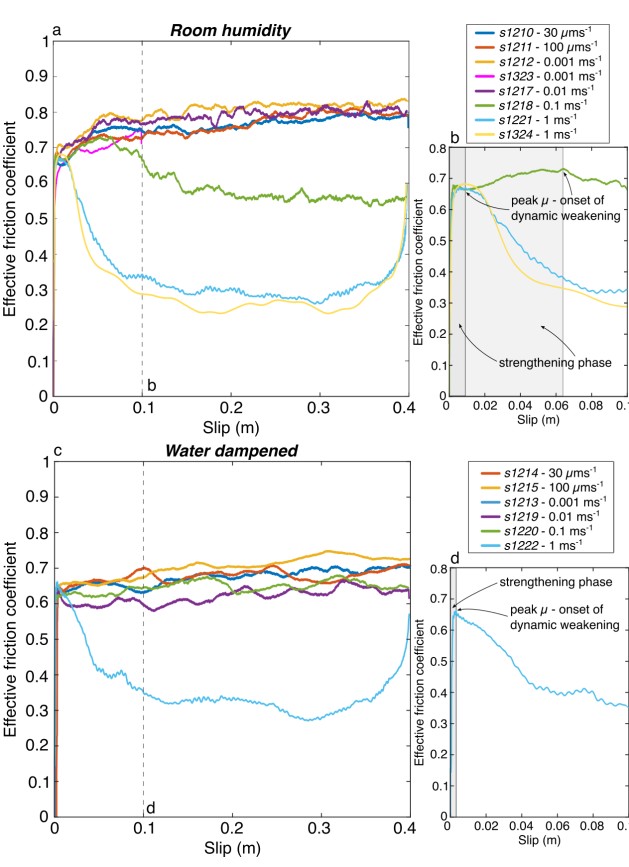

**Figure 2.** Effective friction coefficient in mixed calcite-dolomite gouges. a) and c) Effective friction coefficient versus slip under room humidity and water dampened conditions. c) and d) Detail of effective friction coefficient versus slip in the first 0.1 m of slip in experiments where slip-weakening was observed.



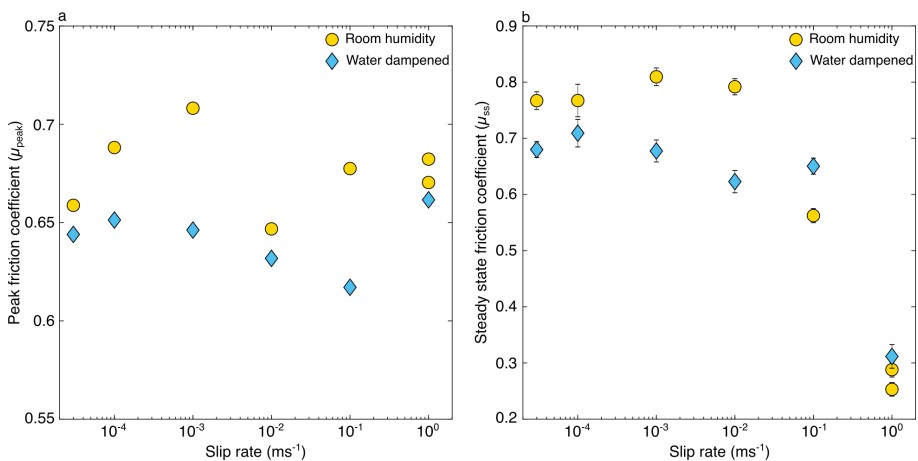

**Figure 3.** Peak and steady state friction coefficient. a) For experiments that showed slip strengthening, the peak friction coefficient was calculated just before the onset of strengthening behaviour in the first 0.1 m of slip. For experiments showing dynamic weakening, the peak friction immediately precedes the friction drop (see Fig. 2b and 2d). b) Steady state friction coefficient was calculated at displacements between 0.15 m and 0.35 m.



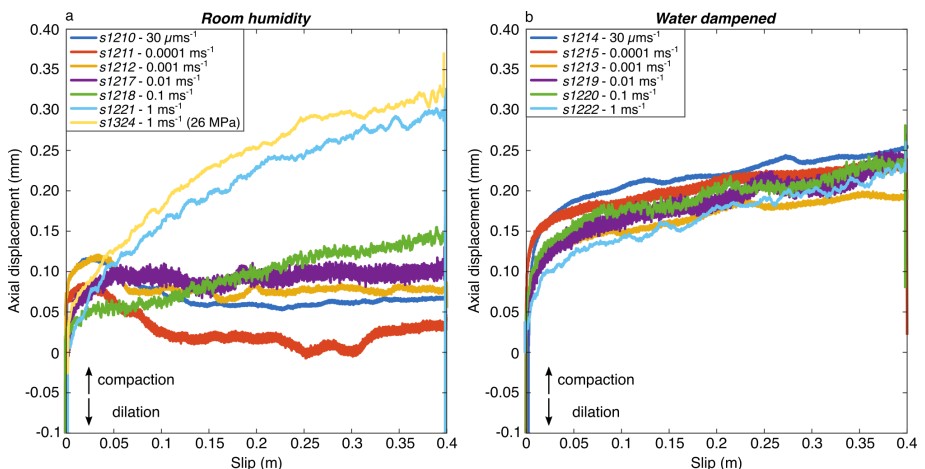

**Figure 4.** Gouge thickness evolution with slip and slip rate. a) Under room humidity conditions and for slip rates of $V \leq 0.01$ ms$^{-1}$, an initial dilation phase (lasting 0.1-0.15 m) was followed by a period of no compaction or dilation was observed. Instead, at higher slip rates ($V \geq 0.1$ ms$^{-1}$), the gouge compacted constantly throughout the whole experiment, with compaction rate increasing with slip rate. b) Under water dampened conditions, the gouge compacted at a similar rate at all investigated slip rates.



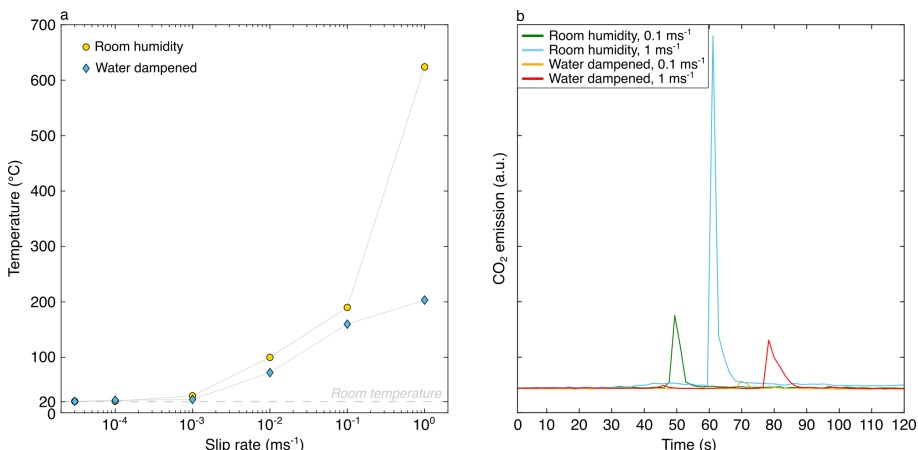

**Figure 5.** Peak temperatures and $CO_2$ emissions. a) Maximum temperature measured by the thermocouple located closest to the gouge layer (see Fig. 1b for location). b) $CO_2$ emissions for experiments at V $\geq$ 0.1 ms$^{-1}$ in both room humidity and water dampened conditions. Greater emissions occur at room humidity conditions, but smaller and distinct peaks are also observed in the presence of water.





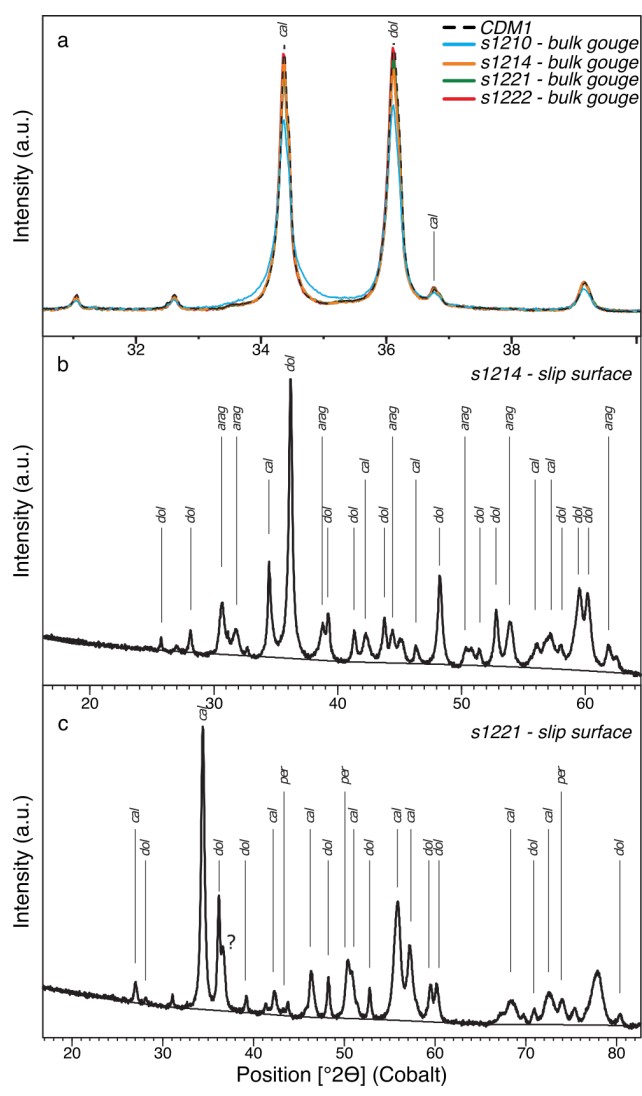

**Figure 6.** XRPD analysis of bulk gouge and slip surfaces. a) Bulk gouge shows a Lorentzian profile for the main calcite peak in experiment *s1210*, suggesting either a large crystallite size distribution or microstrain as a result of intense comminution involving a large fraction of the gouge. b) At 30 $\mu$ms$^{-1}$ in water dampened conditions, traces of aragonite are found on the slip surface as a result of calcite polymorphic transformation during prolonged mechanical grinding. c) For *s1221* (1 ms$^{-1}$ in room humidity conditions), presence of Mg-calcite and periclase (MgO) on the mirror-like slip surface is observed due to decarbonation of dolomite.





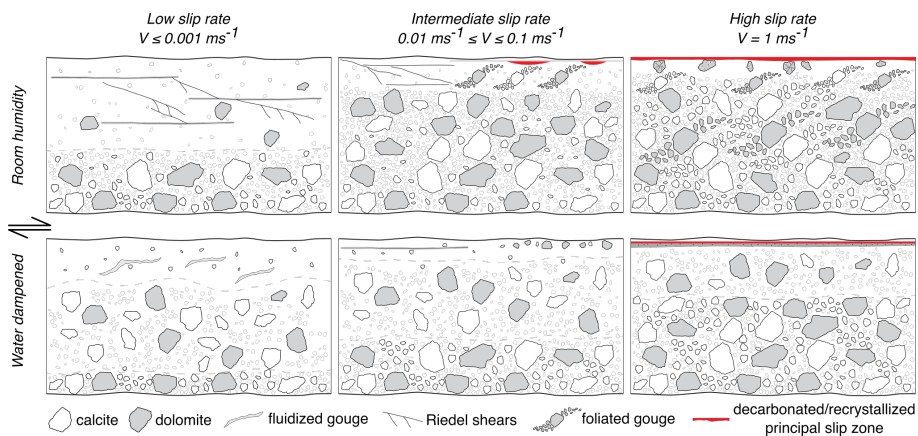

**Figure 7.** Summary of microstructural evolution at different slip rates and deformation conditions.



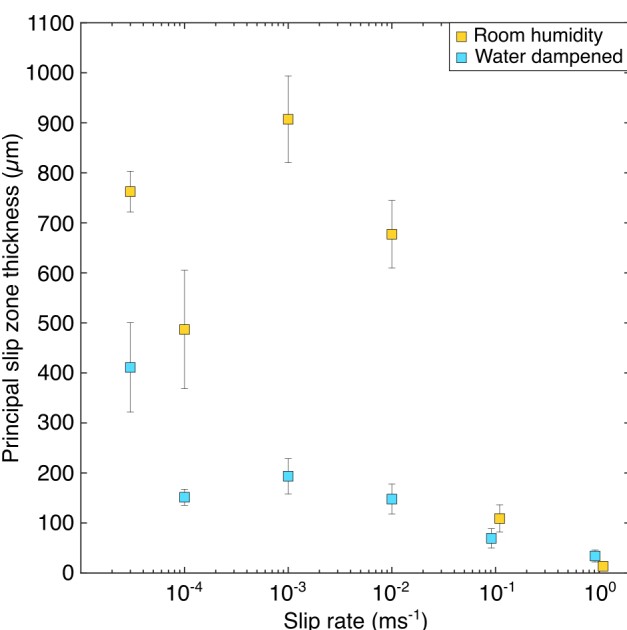

**Figure 8.** Slip zone thickness evolution with slip rate and ambient conditions. The thickness of the localized slip zone decreases almost linearly with log(V) in water dampened experiments. For room humidity conditions, partial sample loss after the experiment means slip zone thickness values are a minimum estimate.





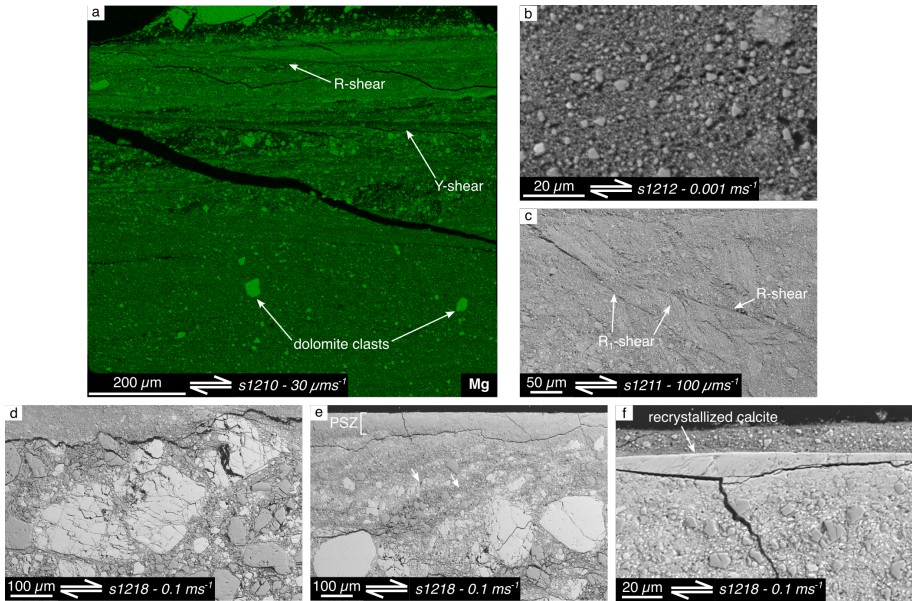

**Figure 9.** Microstructures of experiments in room humidity conditions and V ≤ 0.1 ms$^{-1}$. a) Mg element map of the thick slip zone highlighting the location of remnant dolomite clasts. b) Detail of the fine-grained matrix in the slip zone made of a calcite-dolomite mixture with surviving sub-rounded dolomite clasts up to few tens of micrometres in size. c) The fine-grained slip zone is commonly cut by Y-, R- and R$_1$-shear bands crosscutting each other. d) Enhanced grain size reduction in calcite grains due to fracturing along cleavage planes. e) Development of a weak foliation adjacent to the principal slip zone at V = 0.1 ms$^{-1}$. f) Patches of dynamically recrystallized calcite along the principal slip zone at V = 0.1 ms$^{-1}$.



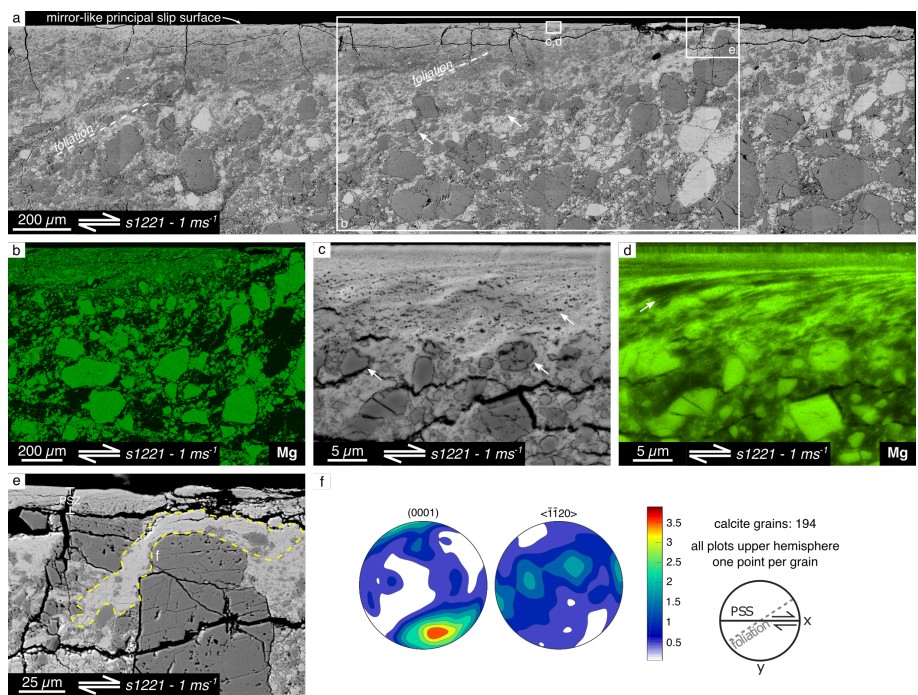

**Figure 10.** Microstructures of experiments in room humidity conditions and V = 1 ms$^{-1}$. a) Development of a foliation consisting in alternation of calcite- and dolomite-rich domains, antithetically inclined c. 40° from the PSS and becoming subparallel to the gouge boundaries when approaching the PSS. Larger dolomite (and when present calcite) clasts have tails of fine-grained material, resembling mantled porphyroclasts. b) Mg element map highlighting foliation development in the bulk gouge. c) Dolomite clasts adjacent to the principal slip zone are characterized by internal holes and vesicular rims interpreted as due to degassing during dolomite decarbonation reaction. Banding of low and higher porosity in the principal slip zone is an indicator for dolomite content. d) Mg element map of c) showing dolomite and calcite banding in the principal slip zone. e) Transition from the principal slip zone to the underlying fractured and foliated gouge with calcite showing evidence of dynamic recrystallization. f) Orientation data for calcite in area highlighted in e). showing the development of a clear CPO along the c-axes.





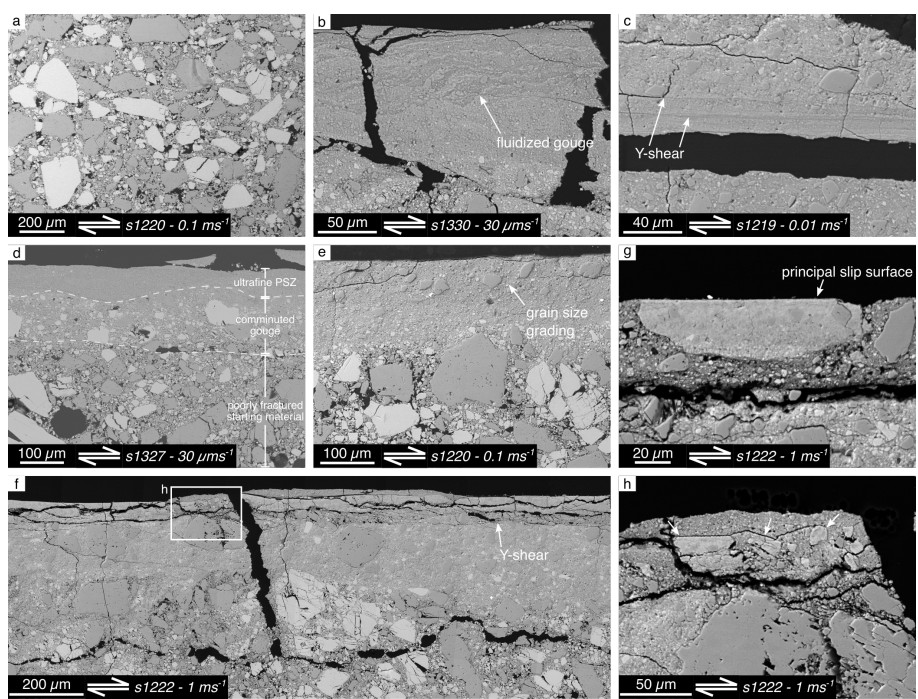

**Figure 11.** Microstructures of experiments in water dampened conditions. a) The bulk gouge is made of poorly fractured calcite and dolomite mixture highly resembling the starting material. b) Occurrence of fluidized structures in the principal slip zone of experiments performed at 30 $\mu$ms$^{-1}$. c) At 0.0001 ms$^{-1}$ ≤ V < 0.1 ms$^{-1}$, the principal slip zone is cut by multiple Y-shears. d) In the presence of fluid water, an ultrafine principal slip surface develops, overlying a highly comminuted gouge which then transitions to a poorly fractured starting material. e) At V = 0.1 ms$^{-1}$, grain size grading is observed in the principal slip zone, with larger clasts occurring near the principal slip surface. f) At V = 1 ms$^{-1}$, strain localizes on a compacted, low porosity, recrystallized slip zone, which g) is not continuous and h) often found broken and reworked.





**Table 1.** Experiments reported in this study.

|  | Experiment | Experimental conditions | Target slip rate ms$^{-1}$ | Displacement m | Normal stress MPa | Mixture batch |
|---|---|---|---|---|---|---|
|  | s1322 | Room humidity | 0.00003 | 0.1 | 17.4 | CDM2 |
|  | s1210 | Room humidity | 0.00003 | 0.4 | 17.4 | CDM1 |
|  | s1211 | Room humidity | 0.0001 | 0.4 | 17.4 | CDM1 |
|  | s1323 | Room humidity | 0.001 | 0.1 | 17.4 | CDM2 |
|  | s1212 | Room humidity | 0.001 | 0.4 | 17.4 | CDM1 |
|  | s1217 | Room humidity | 0.01 | 0.4 | 17.4 | CDM1 |
|  | s1218 | Room humidity | 0.1 | 0.4 | 17.4 | CDM1 |
|  | s1221 | Room humidity | 1 | 0.4 | 17.4 | CDM1 |
|  | s1324 | Room humidity | 1 | 0.4 | 17.4 | CDM2 |
|  |  |  |  |  |  |  |
|  | s1327 | Water dampened | 0.00003 | 0.05 | 17.4 | CDM2 |
|  | s1329 | Water dampened | 0.00003 | 0.1 | 17.4 | CDM2 |
|  | s1328 | Water dampened | 0.00003 | 0.2 | 17.4 | CDM2 |
|  | s1214 | Water dampened | 0.00003 | 0.4 | 17.4 | CDM1 |
|  | s1330 | Water dampened | 0.00003 | 0.4 | 17.4 | CDM2 |
|  | s1215 | Water dampened | 0.0001 | 0.4 | 17.4 | CDM1 |
|  | s1213 | Water dampened | 0.001 | 0.4 | 17.4 | CDM1 |
|  | s1219 | Water dampened | 0.01 | 0.4 | 17.4 | CDM1 |
|  | s1220 | Water dampened | 0.1 | 0.4 | 17.4 | CDM1 |
|  | s1222 | Water dampened | 1 | 0.4 | 17.4 | CDM1 |
|  |  |  |  |  |  |  |
| *Static load* | sld | Room humidity |  |  | 17.4 | CDM1 |
|  | slw | Water dampened |  |  | 17.4 | CDM2 |