# Peer review of "Frictional properties and microstructural evolution of dry and wet calcite-dolomite gouges"

_Solid Earth, 2020_

## Referee Comment (RC1) · B.A. Verberne (Referee) · 20 Oct 2020

**Review of "Frictional properties and microstructural evolution of dry and wet calcite-dolomite gouges" by M. Demurtas et al.**
Submitted to: **Solid Earth**
Manuscript #: **se-2020-170**

**General comments**

The manuscript (ms) by Demurtas et al. reports on an experimental investigation of the frictional behavior of simulated calcite-dolomite faults. Experiments are carried out using the well-known SHIVA rotary shear apparatus, covering a wide range of slip rates ($3 \cdot 10^{-5}$ to 1 m/s) under room humidity and water-dampened conditions (normal stresses 17 to 26 MPa). Much attention is given to post-mortem microstructural analysis, using SEM, EDX, XRD as well as EBSD. The results show marked differences in the frictional strength change with increasing displacement between room humidity and water-dampened samples, specifically in (the velocity dependence of) the onset of dynamic weakening. Based on microstructural evidence, the authors interpret that gouge fluidization played a role in water-dampened experiments, regardless of shearing rate, and that fault rock foliation may develop at co-seismic slip rates.

The authors are well-known for a long list of strong publications in experimental and microstructural fault friction research, and this manuscript seems to be another addition to that. The experiments are technically challenging, but appear to be carried out well using a proven method and a logical, systematic approach. I especially appreciate the emphasis on post-mortem microstructural analyses using a range of different techniques. That said, I do have a few general and technical comments, in particular on the interpretation of EBSD data and inferences on grain size, which I think should be addressed or at least satisfactorily rebutted. I am confident that this ms will come to represent a strong contribution to Solid Earth.

Bart Verberne
20 Oct 2020

**Specific comments**

Lines 27-33: The work summarized here represents shear experiments on gouges as well as compression and/ or torsion tests on dense polycrystals, and even single crystals, deformed under an enormous range of experimental conditions. Elsewhere in the ms (such as in the next paragraph) particular attention is given to a comparison with torsion tests on dense polycrystals. I understand that data on dolomite gouges are sparse, however, I'm not sure if such this comparison is warranted given the major differences in experimental conditions and physical processes at play (in particular the role of porosity and frictional heating). At the very least, a very clear notion of this should be made.
I recall Boneh et al. (2013) published data on dolomite friction, have the authors considered this?

Line 37: Calcite can deform plastically at room temperature, by twinning.

Line 39-40: "For example…..strain localization". Why? How? Some more argumentation is needed here.

Lines 63-65: This interpretation for a CPO formation mechanism is similar to that by Toy et al. 2015, who deserve credit here.

Lines 105-106: Was there pore fluid pressure build-up in the experiment? If not, why not? Some discussion on this, perhaps later on in the ms, where this is explicitly mentioned would be highly topical, I think.

Line 174: Its always a challenge to report microstructural observations in an objective way. While I recognize the importance of Figure 7, I strongly suggest to begin with reporting the observations instead of this highly interpretative sketch. The summarizing sketch should be presented after the key observations have been convincingly demonstrated (i.e., after the present Figs. 8-11).

Line 197: Why not refer to the observations reported in *this* paper (Fig10)? As it is written now, it seems like the reader is referred to other papers by the same authors.

Lines 206-209: Individual grains within the PSZ (which is, I presume, the ultra-comminuted zone immediately adjacent to the PSS as identified by the authors in Fig. 10) are impossible to distinguish from the present BSE images. Furthermore, I contest that the domain highlighted in Fig 10e, and analyzed using EBSD, represents grains from *within* the PSZ. This domain is surrounded by what seem to be coarse fragmented grains, and therefore it is more likely that these represent grains *adjacent* to the PSZ.
At the very least, we need a very clear definition of what represents the PSZ.
The authors indicate that they have prepared thin sections (line 116). Have they tried to image the samples using a polarizing light microscope? This is a cheap and easy way of getting more evidence for a CPO, incl. from within the PSZ, and at a much larger scale than can be achieved using SEM-EBSD (see Verberne et al. 2013, 2019, Niemeijer 2018, Smith et al., 2015). Also, extremely fine grain sizes (<100 nm) are practically impossible to measure, including using t-EBSD.

Lines 265-275: I am puzzled how the authors can connect the grain size observed in the recovered samples to that relevant during the test. If the temperature at the PSS reached up to hundreds of degrees (here 621degC), post-test static recrystallization must have played a role. In fact, with a few simple assumptions for calcite grain growth (which is well constrained), we showed that a grain size of 300-400 nm will be reached within seconds after a test (see Verberne et al. 2017). The implication is that post-test grain size data must be taken with extreme caution. In fact, the grain size likely was much smaller during shear deformation, in the <100 nm range, which has profound implications for their physical properties.
I doubt that the "significant decrease of the activation energy for creep", as claimed in line 273, applies to the grains observed here, or that there is any evidence for this. A dramatic change in the physical properties of nanograins is well-known to occur for grains <100 nm in size, in particular for metals. For calcite, there is evidence for a decrease of the decomposition temperature for a grain size below 50 nm (Wang et al. 2014; see also Verberne et al., 2019). However, the present grains are hundreds of nm's in size, and suggesting that these behave any differently because of their size is a shot in the dark.

Line 287-300: The authors should look at the work of Chen et al. (2017a, b). As I am sure the authors are aware, the presence of water has a buffering effect on temperature, with profound mechanical implications. I think this is highly relevant to the discussion here.

Lines 323-325: Are the authors inferring a low transient permeability during shear deformation? Also, I am curious how this compares with the "dilatancy strengthening" mechanism hypothesized in the 90's (see Marone et al., 1990; Segall and Rice, 1995; Beeler et al., 1996; Samuelson et al., 2009). This may help to broaden the impact of the discussion a bit, I feel. The authors may also consider the role of grain size as well. When grains are extremely tiny, compaction is fast, and transient low permeability or "water-saturated patches" may be readily envisioned.

Line 346: I realize that the authors are probably tired of me pointing this out, but I will continue to express major concerns on claims that MSS's are indicators of co-seismic slip – truncated clasts or not. Although we did never explicitly make a point of this in any of our papers, in my experiments at sub-seismic displacement rates on calcite, I have *also* observed calcite clasts that are truncated by PSZ's (e.g., Fig 8D Verberne et al., Pageoph, or Fig2A Verberne et al, Science). I am not trying to make an of our lives more difficult, but please, reconsider these claims because they are just not consistent with experimental observations.

Lines 366-371: We have also observed this in grains adjacent to the principal slip zone in experiments on calcite gouge at 550°C. Admittedly, the conditions are different, but least the shear rate is closer to what is achieved in here. See Verberne et al. (2017) and Chen et al. (2020).

**Technical corrections/ suggestions**
The manuscript is generally well written.
Throughout the manuscript I noticed an inconsistent use of the hyphen (e.g., water-dampened and water dampened, slip rates and slip-rates, grain size and grain-size). Please check.

Line 7: I suggest to mention the starting layer thickness of 3 mm somewhere within the abstract as well, to be able to put the quoted slip zone width into perspective.
Line 73: "in to" → "into"
Line 153: "cm" → plural
Line 370: I suggest to write out in full what is meant with the range of shear rates here. Mathematically this suggests $10^{0.8}$ s$^{-1}$, which surely is not what is intended.

Discussion
The discussion now consists of two, rather long sections. Perhaps the authors can consider to separate out an 'implications' section, from which the reader can readily take away the more general, geological importance of this work. In my view, one of the nicest results points to the potential role of fluidization over a wide range of slip rates, and on foliation development at co-seismic slip rates.

Figure 8, caption.
"Slip zone thickness evolution with slip rate and ambient conditions". I don't quite follow the last part of this sentence. What is meant by, 'and ambient conditions'?

Figures 9-11
I suggest to print as-large-as-reasonably-possible images at high resolution. It may be the quality of the pre-print that affects the figure quality here, but certainly in Fig 9 there is not an optimal use of space in the rectangular area that is available.

Table 1
Line 106 states that the normal stress was $17.5 \pm 0.1$ MPa. It's a bit strange then, to list 17.4 MPa for each experiment in table 1. Also, I think a typo may have slipped in for experiment s1234. Shouldn't this read 26 MPa?

**References cited which are NOT already listed in the ms:**

Beeler, N. M., Tullis, T. E., Blanpied, M. L., and Weeks, J. D.: Frictional behavior of large displacement experimental faults, J. Geophys. Res., 101, B4, 8697-8715, 1996.

Boneh, Y.; Sagy, A.; Reches, Z. Frictional strength and wear-rate of carbonate faults during high-velocity, steady-state sliding. Earth Planet. Sci. Lett. 2013, 381, 127–137, doi:10.1016/j.epsl.2013.08.050.

Chen, J., Niemeijer A. R, Yao, L., and Ma, S. Water vaporization promotes coseismic fluid pressurization and buffers temperature rise. Geophys. Res. Lett. 44, 5, 2177-2185, 2017.

Chen, J., Niemeijer, A. R., Fokker, P. Vaporization of fault water during seismic slip, Journal of Geophysical Research: Solid Earth, 10.1002/2016JB013824, 122, 6, 4237-4276, 2017.

Chen, J., Verberne, B. A., and Niemeijer, A. R. Flow-to-Friction Transition in Simulated Calcite Gouge: 1 Experiments and Microphysical Modelling. Manuscript accepted for publication in JGR: Solid Earth (per 20 oct 2020).

Delle Piane, C.; Piazolo, S.; Timms, N.; Luzin, V.; Saunders, M.; Bourdet, J.; Giwelli, A.; Ben Clennell, M.; Kong, C.; Rickard, W.; et al. Generation of amorphous carbon and crystallographic texture during low-temperature subseismic slip in calcite fault gouge. Geology 2018, 46, 163–166, doi:10.1130/G39584.1.

Marone, C., Raleigh, C. B., and Scholz, C. H.: Frictional behavior and constitutive modeling of simulated fault gouge, J. Geophys. Res. 95, 7007-7026, 1990.

Niemeijer, A. R. Velocity-dependent slip weakening by the combined operation of pressure solution and foliation development, Sci. Rep. 8, 4724, doi:10.1038/s41598-018-22889-3, 2018.

Samuelson, J. E., Elsworth, D., and Marone, C.: Shear-induced dilatancy of fluid-saturated faults: Experiment and theory, J. Geophys. Res., 114, B12404, doi:10.1029/2008JB006273, 2009.

Segall, P., and Rice, J. R.: Dilatancy, compaction, and slip instability of a fluid-infiltrated fault, J. Geophys. Res., 100, 22155-22171, 1995.

Toy, V.; Mitchell, T.; Druiventak, A.; Wirth, R. Crystallographic preferred orientations may develop in nanocrystalline materials on fault planes due to surface energy interactions. Geochem. Geophys. Geosys. 2015, 16, 2549–2563, doi:10.1002/2015GC005857

Verberne, B. A., Chen, J., Niemeijer, A. R., de Bresser, J. H. P., Pennock, G. M., Drury, M. R., and Spiers, C. J., Microscale cavitation as a mechanism for nucleating earthquakes at the base of the seismogenic zone, Nat. Commun., 8, 1645, doi:10.1038/s41467-017-01843-3, 2017.

Verberne, B. A., de Bresser, J. H. P., Niemeijer, A. R., Spiers, C. J., De Winter, D. A. M., and Plümper, O.: Nanocrystalline slip zones in calcite fault gouge show intense crystallographic preferred orientation: Crystal plasticity at sub-seismic slip rates at 18–150°C, Geology, 41, 863–866, 2013.

Verberne, B. A., Plümper, O., and Spiers, C. J.: Nanocrystalline principal slip zones and their role in controlling crustal fault rheology, Minerals, 9, 328, doi:10.3390/min9060328, 2019.

Verberne, B. A., Plümper, O., De Winter, D. A. M., and Spiers, C. J.: Superplastic nanofibrous slip zones control seismogenic fault friction, Science, 346, 1342–1344, 2014b.

Wang, S.; Cui, Z.; Xia, X.; Xue, Y. Size-dependent decomposition temperature of nanoparticles: A theoretical and experimental study. Physica B 2014, 454, 175–178, doi:10.1016/j.physb.2014.07.058.

---

## Referee Comment (RC2) · John Bedford (Referee) · 4 Dec 2020

General comments

This is my first review of this manuscript. I note that the manuscript has already received publicly available comments from another reviewer, however I chose not to read these comments so that my own thinking on the manuscript was not influenced. I therefore apologize to the authors if I end up repeating comments that have already been made by reviewer 1.

This manuscript investigates the frictional behaviour and microstructural evolution of

mixed calcite-dolomite gouges over a range of slip rates (30 $\mu$m/s - 1 ms) under different water saturation states. As the slip velocity is increased towards coseismic rates the authors document the onset of dynamic weakening and the formation of a defined principal slip zone within the gouge. In room-humidity experiments, at slip velocities >0.1 m/s, they also observe the formation of well-developed foliation in the gouge, something which is commonly observed in natural fault gouges and thought to form by dissolution-precipitation reactions during aseismic creep. The results presented here, where foliations form at coseismic slip rates, suggest that caution should be taken when interpreting the slip history of natural carbonate-bearing faults as similar structures might form over a range of slip velocities.

The manuscript is well written and the results are logically explained. I have made some specific comments below that the authors should address, but this is a useful contribution to the field and I have no issue in recommending it for publication in Solid Earth.

John Bedford (04/12/20)

Specific comments

-l. 29-31: Could the authors be more specific about what they consider to be "low strain rates, high temperatures and high pressures" on line 29, and also "high strain rates, low temperatures and low pressures" on line 31.

-l. 88: Why did the two batches of gouge have different weight percentages?

-l. 106: Is there any reason why a normal stress of 17.5 MPa was chosen for this study? Also why was one test (s1324) ran at a normal stress of 26 MPa? The main goal of the manuscript appears to be to investigate the role of slip rate, displacement and the presence of water on the frictional behaviour and microstructural evolution of calcite-dolomite gouges, therefore it would be sensible to use the same normal stress for all tests in the study. Looking ahead to Figure 2, there is a possible normal stress

dependence on the frictional response (the 26 MPa sample experiences a bit more weakening than the equivalent 17.5 MPa sample), however more than one test under different normal stresses would be required to constrain this relationship. It therefore seems a bit strange to include this test in the manuscript, at least without some further justification in the main text.

-l. 152: It is interesting that adding water makes the gouge compaction slip-rate independent. Do the authors have any explanation for this? Has it been reported in any previous studies?

-l. 162-164 and Fig. 5b: I'm not sure I fully understand this data. It's fine that the $CO_2$ data are qualitative but why are they plotted against time in figure 5b – what is this time relative to? Also why are the $CO_2$ peaks for the fastest experiments (1 m/s) later in time than the slower experiments (0.1 m/s)? I would intuitively expect any thermal decomposition and $CO_2$ release to occur more quickly at faster slip rates.

-l. 174-179: It seems a bit unusual to me that authors include this text, and also present Figures 7 and 8, prior to their detailed microstructural descriptions (and associated figures: 9, 10 & 11) in the following subsections. The authors provide very detailed descriptions of their microstructures in sections 3.5.1 and 3.5.2, with the associated images being presented in figures 9-11. In my opinion it would make more sense to summarize these microstructures and how they differ with slip rate and water saturation (i.e. as shown in Fig. 7) after the detailed descriptions have been presented. In this way the summary figure will "wrap up" the detailed information presented in Figs. 9-11. Perhaps the authors would consider reordering the figures and text slightly?

-l. 224: What is this characteristic wavelength?

-l. 249: I can't see this initial period of dilatancy. Does it occur at the very start of the experiment, at less than 0.01 m of slip? If so it would be good to include an inset in Fig.4 to show this, similar to panels b and d in figure 2.

-l. 286: What temperature does dolomite begin to decompose? This should give a minimum constraint on the temperature rise that occurred in the experiments.

-l. 332-337: Could this discrepancy and low measured temperature rise be a consequence of thermal buffering caused by decomposition of dolomite? As decomposition reactions are generally endothermic they can limit the coseismic temperatures increase, as has been shown for decarbonation reactions (Sulem & Famin, 2009) and dehydration reactions (Brantut et al., 2011) .

Technical corrections

-l. 33: This should read "frictional behaviour of dolomite IS relatively poorly understood".

-l. 228: This should read "with displacements of. . ."

Table 1: Experiment s1324 is listed at a normal stress of 17.4 MPa, but I think this is a typo and should be 26 MPa instead.

References:

Brantut, N., Han, R., Shimamoto, T., Findling, N., & Schubnel, A. (2011). Fast slip with inhibited temperature rise due to mineral dehydration: Evidence from experiments on gypsum. Geology, 39(1), 59–62. https://doi.org/10.1130/g31424.1

Sulem, J., & Famin, V. (2009). Thermal decomposition of carbonates in fault zones: Slip-weakening and temperature-limiting effects. Journal of Geophysical Research: Solid Earth, 114(B3), B03309. https://doi.org/10.1029/2008JB006004

---

## Author Comment (AC1) · 22 Jan 2021

Here below the Reviewer1 (R1) comments are addressed point by point by the authors (A).

[R1] Lines 27-33: The work summarized here represents shear experiments on gouges as well as compression and/ or torsion tests on dense polycrystals, and even single crystals, deformed under an enormous range of experimental conditions. Elsewhere in the ms (such as in the next paragraph) particular attention is given to a comparison with torsion tests on dense polycrystals. I understand that data on dolomite gouges are sparse, however, I'm not sure if such this comparison is warranted given the major differences in experimental conditions and physical processes at play (in particular the role of porosity and frictional heating). At the very least, a very clear notion of this should be made.
I recall Boneh et al. (2013) published data on dolomite friction, have the authors considered this?

[A] We agree with the reviewer comment and we clarified that although the deformation conditions explored in torsion experiments differ significantly to those experienced during shallow faulting, the rheological influence of dolomite on calcite aggregates could be reflected also under more brittle deformation conditions.

We added references for Boneh et al. (2013) and Green et al. (2015) in the studies investigating dolomite friction.

[R1] Line 37: Calcite can deform plastically at room temperature, by twinning.

[A] We agree and modified the phrasing by specifying "calcite can undergo recrystallization…".

[R1] Line 39-40: "For example.....strain localization". Why? How? Some more argumentation is needed here.

[A] We deleted the sentence as it was not adding additional important information to the introduction.

[R1] Lines 63-65: This interpretation for a CPO formation mechanism is similar to that by Toy et al. 2015, who deserve credit here.

[A] Although we agree the both in the case of Toy et al. (2015) and Demurtas et al. (2019a), fracturing is interpreted to occur along weak crystallographic planes, the experimental conditions and processes leading to the CPO development are substantially different. In the case of Toy et al (2015), the CPO is interpreted as a rotation of nanograins immersed in a non-crystalline material due to surface energy interactions in order to maximize coincident site lattices during shearing at 450-600 °C. On the contrary, Demurtas et al. (2019a) infer the CPO to result from mechanically driven grain rotation during granular flow at room temperature.

[R1] Lines 105-106: Was there pore fluid pressure build-up in the experiment? If not, why not? Some discussion on this, perhaps later on in the ms, where this is explicitly mentioned would be highly topical, I think.

[A] Unfortunately the gouge holder used for these experiments does not allow us to measure and control pore pressure. We added a sentence in the Methods section clarifying this aspect.

[R1] Line 174: Its always a challenge to report microstructural observations in an objective way. While I recognize the importance of Figure 7, I strongly suggest to begin with reporting the observations instead of this highly interpretative sketch. The summarizing sketch should be

presented after the key observations have been convincingly demonstrated (i.e., after the present Figs. 8-11).

**[A]** Following both reviewers comments we rearranged the Figures, pushing the old Figure 7 to now Figure 11. We also removed the text that was present at the beginning of section 3.5.

**[R1]** Line 197: Why not refer to the observations reported in *this* paper (Fig10)? As it is written now, it seems like the reader is referred to other papers by the same authors.

**[A]** We added the reference to the figure in this paper.

**[R1]** Lines 206-209: Individual grains within the PSZ (which is, I presume, the ultra-comminuted zone immediately adjacent to the PSS as identified by the authors in Fig. 10) are impossible to distinguish from the present BSE images. Furthermore, I contest that the domain highlighted in Fig 10e, and analyzed using EBSD, represents grains from *within* the PSZ. This domain is surrounded by what seem to be coarse fragmented grains, and therefore it is more likely that these represent grains *adjacent* to the PSZ.

At the very least, we need a very clear definition of what represents the PSZ.
The authors indicate that they have prepared thin sections (line 116). Have they tried to image the samples using a polarizing light microscope? This is a cheap and easy way of getting more evidence for a CPO, incl. from within the PSZ, and at a much larger scale than can be achieved using SEM-EBSD (see Verberne et al. 2013, 2019, Niemeijer 2018, Smith et al., 2015). Also, extremely fine grain sizes (<100 nm) are practically impossible to measure, including using t- EBSD.

**[A]** We agree with the reviewer's comment and changed "within" into "adjacent" as the EBSD analysis in this case has been performed on the recrystallized calcite grains flanking the principal slip zone.

The principal slip zone is already defined and described at lines 305-309: *"The principal slip zone consists of a 15-20 μm thick, extremely fine-grained layer (<<1 μm in size) composed of calcite, Mg-calcite, dolomite, and periclase (EDS and XRPD analysis; Figs. 6 and 9c-d). Calcite forms elongate aggregates with negligible porosity that display an aggregate preferred orientation with the long axes sub-parallel to foliation (Fig. 9c-e)."*

The thickness of the thin sections prepared from the experiments is c. 30-40 μm, which is too thick to image possible CPOs in a polarized microscope with a gypsum plate. During thin section preparation, the difference in strength between calcite and dolomite during thin section polishing made difficult to have a properly even thickness across the whole sample. Therefore, we preferred to have slightly thicker thin section, hence avoiding risking to lose area to the sample due to excessive polishing.

**[R1]** Lines 265-275: I am puzzled how the authors can connect the grain size observed in the recovered samples to that relevant during the test. If the temperature at the PSS reached up to hundreds of degrees (here 621degC), post-test static recrystallization must have played a role. In fact, with a few simple assumptions for calcite grain growth (which is well constrained), we showed that a grain size of 300-400 nm will be reached within seconds after a test (see Verberne et al. 2017). The implication is that post-test grain size data must be taken with extreme caution. In fact, the grain size likely was much smaller during shear deformation, in the <100 nm range, which has profound implications for their physical properties.
I doubt that the "significant decrease of the activation energy for creep", as claimed in line 273,

applies to the grains observed here, or that there is any evidence for this. A dramatic change in the physical properties of nanograins is well-known to occur for grains <100 nm in size, in particular for metals. For calcite, there is evidence for a decrease of the decomposition temperature for a grain size below 50 nm (Wang et al. 2014; see also Verberne et al., 2019). However, the present grains are hundreds of nm's in size, and suggesting that these behave any differently because of their size is a shot in the dark.

[A] In Demurtas et al. (2019, JGR) we interpret the post-experiment microstructures observed in the principal slip zone during TKD analysis, which corresponds to the low porosity, ultrafine grained layer in Figure 9c-e, as fairly representative of that present during deformation. Based on cooling calculation, we assume that annealing was efficient for up to 1 s after the end of the experiment, past which the temperature became too low (<100 °C) to result in significant grain growth. This calculation was highly conservative as the principal slip zone was assumed to be 160 µm thick and made of pure calcite (see Figure 10a in Demurtas et al. (2019, JSG)). In these conditions, we estimated that individual grains may have grown by up to 30–50 nm, resulting in a minimal variation of the grain size distribution (see Figure 4a in Demurtas et al., 2019, JGR). However, grain growth in calcite during annealing is hindered by the presence of impurities (i.e., partial exchange of Ca cations with Mg, see Herwegh et al., 2003, EPSL), as well as pinning of grain boundaries by second phases or pores (e.g., Covey-Crump, 1997, Contr. Min. Petr.; Davis et al., 2011, Phys. Chem. Min.). Grain growth rates in Mg-calcite and dolomite are lower than pure calcite by a factor of $10^3$–$10^4$ (see Davis et al., 2011; Herwegh et al., 2003). XRPD analysis of the principal slip surface (see Figure 6c in the main text) shows a composition of Mg-calcite, dolomite and periclase. Therefore, we believe we can assume that grain growth in the analysed principal slip zone was negligible (on the order of a few nanometres or less), and that the microstructures and grain sizes preserved in our sample are similar to those that were active during the high-velocity experiment.

One aspect that was probably not stressed enough in Demurtas et al. (2019, JGR), is that in the TKD analysis, we miss a layer of c. 1.2 µm from the principal slip surface that was lost during sample preparation. Such thin layers of nanograins (<<100 nm) have been shown to occur in carbonate faults (see e.g., Ohl et al., 2020, EPSL), and could potentially corroborate our idea that change in physical properties of nanograins controlled fault rheology during seismic slip. However, as suggested by the reviewer, the fact that we miss that layer makes the assumption a speculation, which is why in the main text we also provide an alternative interpretation. The real temperature increase in the slip zone was likely to be underestimated due to the large thermal inertia and low acquisition rate (2.5 Hz) of the thermocouples. Aretusini et al. (2019) used optical fibres and detected temperatures c. 300-400 °C higher than those measured with our thermocouples, which, if transferred to our case, would imply temperatures in the principal slip zone of c. 900-1000 °C that could easily explain activation of grain size sensitive creep for the measured grain sizes and shear strain rates.

[R1] Line 287-300: The authors should look at the work of Chen et al. (2017a, b). As I am sure the authors are aware, the presence of water has a buffering effect on temperature, with profound mechanical implications. I think this is highly relevant to the discussion here.

[A] We agree with the reviewer and added the water vaporization as a possible temperature buffer in the discussion section (lines 491-495).

[R1] Lines 323-325: Are the authors inferring a low transient permeability during shear deformation? Also, I am curious how this compares with the "dilatancy strengthening" mechanism hypothesized in the 90's (see Marone et al., 1990; Segall and Rice, 1995; Beeler et al., 1996; Samuelson et al., 2009). This may help to broaden the impact of the discussion a bit, I feel. The authors may also consider the

role of grain size as well. When grains are extremely tiny, compaction is fast, and transient low permeability or "water-saturated patches" may be readily envisioned.

**[A]** We agree with the reviewer. Fast grain size reduction, which is more efficient in calcite and testified by the presence of only dolomite surviving grains (see also Figures 5.12 and 5.13 in Smith et al., 2017), can create local compositionally heterogeneous patches within the principal slip zone. Areas richer in calcite would likely have a slightly lower grain size, and represent regions of lower permeability in the principal slip zone, allowing for local build-up of the pore pressure. Once this pressure is released, the gouge fluidizes and the materials is injected in the adjacent higher permeable gouge. However, these pore pressure oscillations are likely to be minimal and occur only at a local level, since there is no clear signal of abrupt compaction/dilation during deformation.

Regarding the "dilatancy strengthening" mechanism: for experiments run at $V \leq 0.01$ ms$^{-1}$ and room-humidity conditions, we show that the slip strengthening behaviour of the gouge is associated with dilation (Fig. 4a) and development of a thick (>500 µm) layer of distributed deformation. These observations nicely compare in fact with the "dilatancy strengthening" mechanism suggested by e.g., Marone et al. (1990) and Beeler et al. (1996). However, unlikely the work from Samuelson et al. (2009), in our experiments dilatancy is not observed under water-dampened conditions (Fig. 4b), but rather the gouges compact at a similar rate at all investigated slip rates and the mechanical data show only slight slip strengthening to slip neutral behaviour for slip rates up to 0.1 ms$^{-1}$. Results similar to ours have also been documented by Rempe et al. (2017, 2020) in rotary shear experiments on wet calcite gouges, with both the same gouge holder than ours, and with a dedicated gouge holder allowing to run experiments under pressure- or volume-control.

**[R1]** Line 346: I realize that the authors are probably tired of me pointing this out, but I will continue to express major concerns on claims that MSS's are indicators of co-seismic slip – truncated clasts or not. Although we did never explicitly make a point of this in any of our papers, in my experiments at sub-seismic displacement rates on calcite, I have *also* observed calcite clasts that are truncated by PSZ's (e.g., Fig 8D Verberne et al., Pageoph, or Fig2A Verberne et al, Science). I am not trying to make an of our lives more difficult, but please, reconsider these claims because they are just not consistent with experimental observations.

**[A]** We deleted the reference to mirror surfaces and replaced with *"slip zones containing recrystallized material"*.

**[R1]** Lines 366-371: We have also observed this in grains adjacent to the principal slip zone in experiments on calcite gouge at 550°C. Admittedly, the conditions are different, but least the shear rate is closer to what is achieved in here. See Verberne et al. (2017) and Chen et al. (2020).

**[A]** Although there is indeed some grain elongation in the work of Chen et al. (2020), we struggle to see the development of a well-defined foliation, which is the main point of this section. The inclined bands shown e.g., in Fig. 5 of Chen et al. (2020) seem to be better interpreted as Riedel shear bands (their orientation also matches) rather than foliation. Regarding the microstructures shown in Verberne et al. (2017), we see the point why grain elongation and porosity alignment in Fig. 3 could be interpreted as a slight foliation. We added reference to Verberne et al. (2017) in the main text.

**[R1]** Throughout the manuscript I noticed an inconsistent use of the hyphen (e.g., water-dampened and water dampened, slip rates and slip-rates, grain size and grain-size). Please check.

**[A]** We checked and corrected the use of the hyphen throughout the main text.

**[R1]** Line 7: I suggest to mention the starting layer thickness of 3 mm somewhere within the abstract as well, to be able to put the quoted slip zone width into perspective.

**[A]** We added the initial gouge thickness in the Abstract.

**[R1]** Line 73: "in to" -> "into"

**[A]** Corrected.

**[R1]** Line 153: "cm" -> plural

**[A]** The symbol "cm" can be used both as singular and plural.

**[R1]** Line 370: I suggest to write out in full what is meant with the range of shear rates here. Mathematically this suggests $10^{0.8}$ s$^{-1}$ , which surely is not what is intended.

**[A]** We corrected the notation in $10^{-4}$-$10^{-5}$ s$^{-1}$.

**[R1]** The discussion now consists of two, rather long sections. Perhaps the authors can consider to separate out an 'implications' section, from which the reader can readily take away the more general, geological importance of this work. In my view, one of the nicest results points to the potential role of fluidization over a wide range of slip rates, and on foliation development at co-seismic slip rates.

**[A]** We rearranged the Discussion section and separated a "Implication for natural fault zones" part describing the occurrence of fluidized structures across a wide range of slip rates and occurrence of foliated cataclasites.

**[R1]** Figure 8, caption.
"Slip zone thickness evolution with slip rate and ambient conditions". I don't quite follow the last part of this sentence. What is meant by, 'and ambient conditions'?

**[A]** With ambient conditions we meant presence or absence of fluid water (room-dry vs. water-dampened conditions). We changed the caption in "Slip zone thickness evolution with slip rate and presence of water".

**[R1]** Figures 9-11
I suggest to print as-large-as-reasonably-possible images at high resolution. It may be the quality of the pre-print that affects the figure quality here, but certainly in Fig 9 there is not an optimal use of space in the rectangular area that is available.

**[A]** We expanded the size of Figures with the microstructural observations.

**[R1]** Table 1
Line 106 states that the normal stress was 17.5 ± 0.1 MPa. It's a bit strange then, to list 17.4 MPa for each experiment in table 1. Also, I think a typo may have slipped in for experiment s1234. Shouldn't this read 26 MPa?

**[A]** Corrected.

---

## Author Comment (AC2) · 22 Jan 2021

Here below the Reviewer2 (R2) comments are addressed point by point by the authors (A).

[R2] -l. 29-31: Could the authors be more specific about what they consider to be "low strain rates, high temperatures and high pressures" on line 29, and also "high strain rates, low temperatures and low pressures" on line 31.

[A] We specified the ranges of strain/shear rates, temperature and pressure for the cases listed.

[R2] -l. 88: Why did the two batches of gouge have different weight percentages?

[A] The two batches of calcite-dolomite gouge were made my weighting the equal amount of calcite and dolomite and then mixing it together. The difference in the calcite-dolomite ratio between the two batches could be due to little manual mixing of the gouge before the mineralogical analysis, or sampling of a slightly dolomite-richer portion.

[R2] -l. 106: Is there any reason why a normal stress of 17.5 MPa was chosen for this study? Also why was one test (s1324) ran at a normal stress of 26 MPa? The main goal of the manuscript appears to be to investigate the role of slip rate, displacement and the presence of water on the frictional behaviour and microstructural evolution of calcite-dolomite gouges, therefore it would be sensible to use the same normal stress for all tests in the study. Looking ahead to Figure 2, there is a possible normal stress dependence on the frictional response (the 26 MPa sample experiences a bit more weakening than the equivalent 17.5 MPa sample), however more than one test under different normal stresses would be required to constrain this relationship. It therefore seems a bit strange to include this test in the manuscript, at least without some further justification in the main text.

[A] Since one of the aims of our work was to further explore the microstructural evolution under different deformation conditions starting from the work of Smith et al. (2017), we chose to perform the experiments at a very similar normal stress (17.3 in Smith et al., 2017 vs. 17.5 in our study) in order to minimize the change of variables.

We agree with the reviewer and since only one experiment was carried out at 26 MPa, we decided it exclude it from the dataset and focus mainly on the influence of slip rate, displacement and presence of water.

[R2] -l. 152: It is interesting that adding water makes the gouge compaction slip-rate independent. Do the authors have any explanation for this? Has it been reported in any previous studies?

[A] As pointed out by Reviewer1, the creation of a very fine-grained (<<1 µm) slip zone in all water-dampened experiments which varies only in thickness with increasing slip rate, facilitates and promotes compaction during deformation. Although this can explain the compaction in the presence of water, it still poses a question of why same compaction history seems to be independent from slip rate, question to which we currently have no answer to.

Rempe et al. (2017, JSG; 2020, JGR) performed rotary shear experiments at slip rates of 0.001 ms$^{-1}$ and 1 ms$^{-1}$ on calcite gouges in both water-dampened conditions (using the same gouge holder as us) and water-saturated conditions with either fluid pressure- or volume-control. Their axial displacement data show that for the range of displacements investigated in our study (<0.4 m) the gouge thickness evolution appears to be very similar in the presence of water and independent of the drainage conditions. At displacements >0.5 m (which is past the displacement explored in our work, see e.g., Figure 4e in Rempe et al., 2020), dilation occurs in undrained experiments and is

interpreted as pore fluid pressurization due to temperature increase in the principal slip zone during dynamic weakening.

[R2] -l. 162-164 and Fig. 5b: I'm not sure I fully understand this data. It's fine that the CO2 data are qualitative but why are they plotted against time in figure 5b – what is this time relative to? Also why are the CO2 peaks for the fastest experiments (1 m/s) later in time than the slower experiments (0.1 m/s)? I would intuitively expect any thermal decomposition and CO2 release to occur more quickly at faster slip rates.

[A] The gas emissions were recorded with a separate control system from the rotary shear apparatus. This means that the measuring had to be started manually and that makes it unable for us to constrain gas emission data with those collected with SHIVA (e.g., friction, slip, T). Therefore, data are plotted in time as they were recorded. The shift in time of the $CO_2$ peaks is due to the fact the measurements were started at different times (few seconds of difference) from the start of the experiments. We added a few lines in the main text to clarify this point.

*"Since the acquisition system for gas emissions was separated and not synchronized with that of SHIVA, $CO_2$ emissions are plotted against time. Shifts in time of the peaks from different experiments is due to changes in the manual start of the data collection."*

[R2] -l. 174-179: It seems a bit unusual to me that authors include this text, and also present Figures 7 and 8, prior to their detailed microstructural descriptions (and associated figures: 9, 10 & 11) in the following subsections. The authors provide very detailed descriptions of their microstructures in sections 3.5.1 and 3.5.2, with the associated images being presented in figures 9-11. In my opinion it would make more sense to summarize these microstructures and how they differ with slip rate and water saturation (i.e. as shown in Fig. 7) after the detailed descriptions have been presented. In this way the summary figure will "wrap up" the detailed information presented in Figs. 9-11. Perhaps the authors would consider reordering the figures and text slightly?

[A] Following both reviewers comments we rearranged the Figures, pushing the old Figure 7 to now Figure 11. We also removed the text that was present at the beginning of section 3.5.

[R2] -l. 224: What is this characteristic wavelength?

[A] We added the wavelength (i.e., c. 300 µm) of the boundary between the two microstructural domains in the main text.

[R2] -l. 249: I can't see this initial period of dilatancy. Does it occur at the very start of the experiment, at less than 0.01 m of slip? If so it would be good to include an inset in Fig.4 to show this, similar to panels b and d in figure 2.

[A] We added an inset in Fig. 4a showing the initial dilatancy for experiment *s1221* performed at 1 ms$^{-1}$ and modified the figure caption.

[R2] -l. 286: What temperature does dolomite begin to decompose? This should give a minimum constraint on the temperature rise that occurred in the experiments.

[A] The temperature for the start of dolomite decarbonation is c. 550 °C and was presented in the Results section when describing the variation in mineralogy of the gouges. We added it again in the Discussion.

**[R2]** -l. 332-337: Could this discrepancy and low measured temperature rise be a consequence of thermal buffering caused by decomposition of dolomite? As decomposition reactions are generally endothermic they can limit the coseismic temperatures increase, as has been shown for decarbonation reactions (Sulem & Famin, 2009) and dehydration reactions (Brantut et al., 2011) .

**[A]** $CO_2$ emissions were detected under both room-humidity and water-dampened conditions. Therefore, a possible buffer effect of the decomposition reaction would likely not result in striking differences between the two cases. However, as noted by Reviewer 1, the occurrence of water could be an alternative, or partial, explanation of the lower temperatures observed due to a buffer effect during vaporization. We have added this part into the discussion section.

**[R2]** -l. 33: This should read "frictional behaviour of dolomite IS relatively poorly under- stood".

**[A]** Corrected.

**[R2]** -l. 228: This should read "with displacements of. . ."

**[A]** Corrected.

**[R2]** Table 1: Experiment s1324 is listed at a normal stress of 17.4 MPa, but I think this is a typo and should be 26 MPa instead.

**[A]** Corrected.